# Investigating Mixture Policies in Entropy-Regularized Actor-Critic

## Abstract

We study mixture policies in entropy-regularized reinforcement learning. Mixture policies offer greater flexibility than base policies like Gaussians, which we show theoretically provides improved solution quality and robustness to the entropy scale. Despite these potential benefits, they are rarely used for algorithms like Soft Actor-Critic, potentially due to the fact that Gaussians are easily reparameterized to get lower variance gradient updates, but mixtures are not. We fill this gap, introducing reparameterization gradient estimators for the mixture policy. Through extensive experiments on environments from classic control, MuJoCo, the DeepMind Control Suite and a suite of randomly generated bandits, our results show that mixture policies explore more efficiently in tasks with unshaped rewards (across entropy scales), while performing comparably to base policies in tasks with shaped rewards, and are more robust to multimodal critic surfaces.

## 1 Introduction

Policy gradient methods are widely used in online reinforcement learning (RL), particularly for continuous action spaces. However, there are many design decisions in these methods that remain under-explored. One of these is the choice of policy parameterization. Gaussian policies are by far the most common parameterization (Williams, 1992; Degris et al., 2012; Lillicrap et al., 2015; Duan et al., 2016; Schulman et al., 2017; Neumann et al., 2022), or its bounded variants like squashed Gaussian policies (Haarnoja et al., 2018a). There are a handful of works exploring other distributions, including beta policies (Chou et al., 2017) and the family of heavy-tailed policies (Kobayashi, 2019; Bedi et al., 2024). Using mixture policies, such as a conditional Gaussian mixture model for the policy, however, has been largely unexplored.

Yet there are reasons that this increased flexibility from mixture policies could be beneficial. A more flexible policy class may contain better optimal policies. For example, when the environment is partially observable, the optimal policy could be stochastic and multimodal. Even in fully observable settings, it is now common to use entropy-regularized objectives, which prefer stochastic policies; multimodal rather than unimodal policies may be better in this regime. The increased flexibility may also facilitate exploration. Several works have shown that heavy-tailed policies may improve learning through persistent exploration compared to Gaussian policies (Kobayashi, 2019; Bedi et al., 2024). Mixture policies have the potential to provide mode-directed exploration, by keeping probability high for multiple promising actions.

Though under-explored, some work has looked at more flexible policy classes. Implicit policies use deep generative models (e.g., energy-based models (Haarnoja et al., 2017; Messaoud et al., 2024); normalizing flows (Tang & Agrawal, 2018; Mazoure et al., 2020); diffusion models (Wang et al., 2023)) for the policy. Compared to these more complex distributions, policies using parametric distributions like mixture models have two benefits: they are simpler to train and the explicit densities are useful in entropy-regularized RL. Otherwise, several unpublished works briefly touch on mixture policies. The first version of the SAC paper (Haarnoja et al., 2018b) did test mixture policies but then did not pursue this further nor provide insights on this choice. We hypothesize the lack of reparameterization estimators for mixture policies might be the reason that later versions of SAC switched to a single Gaussian, as they found the reparameterization gradient estimator works better (see Footnote 3 on Page 67 of Haarnoja, 2018). Later, Hou et al. (2020) tried to avoid reparameterization of the whole mixture policy in SAC by using a separate objective for the weighting policy,

but they found little improvement from their approach. Finally, Baram et al. (2021) explore the utility of the upper and lower bounds of the mixture model's entropy, without considering a learnable weighting policy. We provide a more in-depth discussion on related work in Appendix A.

In this work, we provide a directed study on the potential benefits of mixture policies in the entropy-regularized setting. We first prove that mixture models provide improved solution quality: they have comparable or better objective values and are more robust to larger entropy regularization, in that stationary points may not exist for the Gaussian policy but do for the Gaussian mixture. We then derive two reparameterized gradient updates: a partial reparameterization obtained by extending the reparameterization policy gradient theorem (Lan et al., 2022) and a reparameterization combining the Gumbel-softmax reparameterization with the reparameterization on the base policy in the mixture. Though simple, to the best of our knowledge, these reparameterizations have not been proposed for conditional mixture models. We test mixture policies and these two reparameterization updates with Soft-Actor Critic (SAC; Haarnoja et al., 2018a) in seven MuJoCo, twelve DeepMind Control Suite and three classic control environments, and note that differences only arise for environments without shaped rewards. For such uninformative rewards, that do not guide the agent to the goal, we find the mixture policies more consistently find the goal and perform better for a wider range of entropy scales. We find that the critic is less smooth, with more peaks, in the unshaped setting, and that the policy actually uses multiple modes for such a critic. Further, in targeted experiments in multimodal bandits to mimic such a multimodal critic, we find the mixture policy more often finds the maximal peak compared to the base policy.

## 2 PROBLEM FORMULATION

We consider the standard Markov decision process (MDP) problem setting. An MDP is defined by $\langle \mathcal{S}, \mathcal{A}, p, d_0, r, \gamma \rangle$, where $\mathcal{S}$ is the state space, $\mathcal{A}$ is the action space, $p$ is the transition function, $d_0$ is the initial state distribution, $r$ is the reward function, and $\gamma$ is the discount factor. In this paper, we consider $\mathcal{A}$ to be continuous, and $r$ is deterministic and bounded by $[-r_{\max}, r_{\max}]$. Define $\mathbb{E}_\pi[\sum_{t=0}^\infty \cdot] \doteq \mathbb{E}_{S_0 \sim d_0, A_t \sim \pi(\cdot|S_t), S_{t+1} \sim p(\cdot|S_{t-1}, A_{t-1})}[\sum_{t=0}^\infty \cdot]$, the agent's goal is to find a policy $\pi$ that maximizes the *expected return* from the start states:

$$J_0(\pi) \doteq \mathbb{E}_\pi\big[\sum_{t=0}^\infty \gamma^t r(S_t, A_t)\big]. \tag{1}$$

Oftentimes, the agent optimizes the *entropy-regularized objective* that promotes stochastic policies:

$$J(\pi) \doteq \mathbb{E}_\pi\big[\sum_{t=0}^\infty \gamma^t\big(r(S_t, A_t) + \alpha \mathcal{H}(\pi(\cdot|S_t))\big)\big] = \mathbb{E}_{s \sim d_0, a \sim \pi(\cdot|s)}\left[Q_\pi(s, a) - \alpha \log \pi(a|s)\right], \tag{2}$$

where $\alpha$ is the entropy scale, $\mathcal{H}(q) \doteq -\int q(x) \log q(x)\, dx$ is the differential entropy for distribution $q(x)$, and $Q_\pi(s, a) \doteq \mathbb{E}_\pi[\sum_{t=0}^\infty \gamma^t(r(S_t, A_t) + \alpha\gamma\mathcal{H}(\pi(\cdot|S_{t+1})))]$ is the soft action-value function.

Soft Actor-Critic (SAC; Haarnoja et al., 2018a) learns $\pi$ through maximizing a surrogate of (2):

$$\hat{J}(\pi_{\boldsymbol{\theta}}) = \mathbb{E}_{S_t \sim \mathcal{B}, A_t \sim \pi_{\boldsymbol{\theta}}(\cdot|S_t)}\left[Q_{\mathbf{w}}(S_t, A_t) - \alpha \log \pi_{\boldsymbol{\theta}}(A_t|S_t)\right]. \tag{3}$$

where $\mathcal{B}$ is a buffer of collected data and $Q_{\mathbf{w}}$ is an estimate of $Q_{\pi_{\boldsymbol{\theta}}}$. In this work, we focus on the role of policy parameterization; we refer the reader to the original paper for other details on SAC.

We can obtain an unbiased sample of the gradient of (3) in two ways. One is the *likelihood-ratio* gradient estimator (Williams, 1992):

$$\hat{\nabla}_{\boldsymbol{\theta}}\hat{J}(\pi_{\boldsymbol{\theta}}) = \nabla_{\boldsymbol{\theta}} \log \pi_{\boldsymbol{\theta}}(A_t|S_t)\big(Q_{\mathbf{w}}(S_t, A_t) - \alpha \log \pi_{\boldsymbol{\theta}}(A_t|S_t)\big). \tag{4}$$

The likelihood-ratio estimator often suffers from high variance, and a baseline is used to reduce variance. When the action is reparameterizable, for example, $A_t = f_{\boldsymbol{\theta}}(\epsilon_t; S_t)$, an alternative is to use the *reparameterization* gradient estimator:

$$\hat{\nabla}_{\boldsymbol{\theta}}\hat{J}(\pi_{\boldsymbol{\theta}}) = \nabla_{\boldsymbol{\theta}}\big(Q_{\mathbf{w}}(S_t, f_{\boldsymbol{\theta}}(\epsilon_t; S_t)) - \alpha \log \pi_{\boldsymbol{\theta}}(f_{\boldsymbol{\theta}}(\epsilon_t; S_t)|S_t)\big), \tag{5}$$

where $\epsilon_t$ is a sample from a noise distribution $p(\cdot)$. Though there is no guarantee that the reparameterization estimator is better than the likelihood-ratio estimator in general (Gal, 2016; Parmas et al., 2018), it is shown to have lower variance under some assumptions (Xu et al., 2019).

*Gaussian policies* are a common choice when the action space is continuous:

$$\pi_{\boldsymbol{\theta}}(a|s) = \mathcal{N}(a; \mu_{\boldsymbol{\theta}}(s), \sigma_{\boldsymbol{\theta}}(s)^2), \tag{6}$$

where $\mu_{\boldsymbol{\theta}}(s)$ is the mean and $\sigma_{\boldsymbol{\theta}}(s)$ is the standard deviation. Gaussian policies have infinite support, but the action space is typically bounded in practice. To address the bias of truncating the density, *squashed Gaussian policies* use $\tanh$ to transform the unbounded support to a bounded interval:

$$\pi_{\boldsymbol{\theta}}(a|s) = \mathcal{N}(\tanh^{-1}(a); \mu_{\boldsymbol{\theta}}(s), \sigma_{\boldsymbol{\theta}}(s)^2). \tag{7}$$

In this paper, we study *mixture policies* with $N \in \mathbb{N}^+$ components:

$$\pi_{\boldsymbol{\theta}}^m(a|s) = \sum_{k=1}^N \pi_{\boldsymbol{\theta}}^w(k|s)\pi_{\boldsymbol{\theta}}^b(a|s,k),$$

where $\pi_{\boldsymbol{\theta}}^w$ is the *weighting policy* and $\pi_{\boldsymbol{\theta}}^b$ with different $k$ are the *component policies*. When needed, we also explicitly write $\pi_{\boldsymbol{\theta}^m}^m(a|s) = \sum_{k=1}^N \pi_{\boldsymbol{\theta}^w}^w(k|s)\pi_{\boldsymbol{\theta}_k^b}^b(a|s)$, where $\boldsymbol{\theta}^m = \left[\boldsymbol{\theta}_1^{b\top}, \cdots, \boldsymbol{\theta}_N^{b\top}, \boldsymbol{\theta}^{w\top}\right]^\top$. The weighting policy is usually parameterized as a softmax policy, while the component policies can be any continuous policy. When the component policies are Gaussian policies (6), we call the resulting policy the *Gaussian mixture (GM) policy*. In this context, we call the Gaussian policy the *base policy*. Similarly, when the base policy is the squashed Gaussian policy (7), we call the resulting mixture policy the *squashed Gaussian mixture (SGM) policy*.

# 3 ROBUSTNESS OF MIXTURE POLICIES TO ENTROPY REGULARIZATION

The policy parameterization influences the set of stationary points of the entropy-regularized objective in (3), which is non-concave (Agarwal et al., 2019). We first show that mixture policies improve solution quality, in terms of having a higher regularized objective and a higher unregularized objective if an entropy-constrained optimization is used. Then we show that mixtures are robust to higher levels of entropy, both theoretically and empirically. Appendix B contains proofs for this section.

## 3.1 OPTIMALITY OF STATIONARY POINTS

We first show that the optimal stationary points, namely $\boldsymbol{\theta}^* \doteq \arg\max_{\boldsymbol{\theta} \in \{\boldsymbol{\theta} | \nabla_{\boldsymbol{\theta}} J(\pi_{\boldsymbol{\theta}}) = \mathbf{0}\}} J(\pi_{\boldsymbol{\theta}})$, of the mixture policy is at least as good as or better than the base policy in Proposition 3.1.

**Proposition 3.1.** *When both $\pi_{\boldsymbol{\theta}^{b,*}}^b$ and $\pi_{\boldsymbol{\theta}^{m,*}}^m$ exist, then $J(\pi_{\boldsymbol{\theta}^{m,*}}^m) \geq J(\pi_{\boldsymbol{\theta}^{b,*}}^b)$.*

The inequality is likely strict when the return landscape is multimodal, as the mixture policy can maintain high returns while increasing its entropy by splitting its density into different modes (see Figure 1 (Left) for intuition).

The next natural question is how their optimal stationary points compare regarding the unregularized objective (the expected return) $J_0(\pi_{\boldsymbol{\theta}})$. In general, it is difficult to guarantee $J_0(\pi_{\boldsymbol{\theta}^{m,*}}^m) \geq J_0(\pi_{\boldsymbol{\theta}^{b,*}}^b)$. However, when the entropy is imposed as a constraint instead of regularization, the mixture policy is guaranteed to be at least as good as the base policy, as shown in Proposition 3.2.

**Proposition 3.2.** *Consider entropy-constrained policy optimization $\max_{\boldsymbol{\theta}} J_0(\pi_{\boldsymbol{\theta}})$ subject to $\mathcal{H}(\pi_{\boldsymbol{\theta}}) \geq H$ for some $H > 0$ and define the optimal solution as $\boldsymbol{\theta}'$, then $J_0(\pi_{\boldsymbol{\theta}^{m,\prime}}^m) \geq J_0(\pi_{\boldsymbol{\theta}^{b,\prime}}^b)$.*

## 3.2 NON-EXISTENCE OF STATIONARY POINTS WITH STRONG ENTROPY REGULARIZATION

This section shows that the mixture policy may have stationary points in scenarios where the base policy does not. We focus on the bandit setting, in which case, the objective degenerates to

$$J(\pi_{\boldsymbol{\theta}}) = \mathbb{E}_{a \sim \pi_{\boldsymbol{\theta}}}[r(a) - \alpha \ln \pi_{\boldsymbol{\theta}}(a)]. \tag{8}$$

Proposition 3.3 shows that for a sufficiently large entropy regularization, there are no stationary points for the Gaussian policy.

**Proposition 3.3.** *Assume $r : \mathcal{A} \to \mathbb{R}$ is an integrable function on $\mathcal{A} = \mathbb{R}$. For all $\alpha > \frac{3}{2}r_{\max}$, $J(\pi_{\mu,\sigma}) = \mathbb{E}_{a \sim \mathcal{N}(\mu,\sigma)}[r(a) - \alpha \log \mathcal{N}(a; \mu, \sigma)]$ does not have any stationary point.*

The Gaussian mixture (GM) policy, on the other hand, is less sensitive to entropy regularization. Specifically, we show that for every stationary point of the regularized objective with the base policy, there exists a corresponding set of stationary points for the mixture policy.

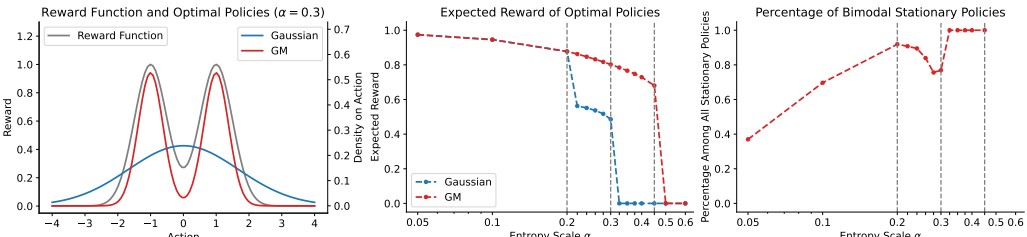

Figure 1: Numerical study on a bimodal bandit. Left: Reward function of the bandit and optimal policies when $\alpha = 0.3$. Middle: Expected reward of the optimal policies for different $\alpha$. Right: The percentage of bimodal policies among all stationary GM policies found in 100 trials. The dashed vertical lines in the figure mark different levels of entropy regularization, each having qualitatively distinct effects on the Gaussian and GM policies.

**Proposition 3.4.** *For any $\tilde{\boldsymbol{\theta}}^b$ such that $\nabla_{\boldsymbol{\theta}^b} J(\pi^b_{\tilde{\boldsymbol{\theta}}^b}) = 0$ and arbitrary $\boldsymbol{\theta}^w$, we have $\nabla_{\boldsymbol{\theta}^m} J(\pi^m_{\tilde{\boldsymbol{\theta}}^m}) = 0$, where $\tilde{\boldsymbol{\theta}}^m = \left[ \tilde{\boldsymbol{\theta}}^{b\top}, \cdots, \tilde{\boldsymbol{\theta}}^{b\top}, \boldsymbol{\theta}^{w\top} \right]^\top$.*

Following from Proposition 3.4, we have the following remark: the minimum $\alpha$ after which the mixture policy does not have a stationary point is at least as large as the base policy.

*Remark 3.5.* Define $\alpha^\pi_{\min} = \inf\{\alpha \mid \nabla_{\boldsymbol{\theta}} J(\pi_{\boldsymbol{\theta}}) \neq \mathbf{0}, \forall \boldsymbol{\theta}\}$ for policy $\pi_{\boldsymbol{\theta}}$, then $\alpha^{\pi^m}_{\min} \geq \alpha^{\pi^b}_{\min}$.

In fact, it is possible to find examples where the inequality is strict. Figure 1 (Left) shows such an example. In this example with $\alpha = 0.3$, the optimal GM policy has two separate modes covering two modes of the reward function, while the optimal Gaussian policy has its only mode covering the the middle of the two reward modes. When $\alpha = 0.4$, the Gaussian policy's standard deviation always diverges to infinity, while the GM policy still has good stationary points.

### 3.3 A NUMERICAL STUDY ON A BIMODAL BANDIT

In this section, we empirically corroborate that mixtures are more robustness to entropy regularization and better balance entropy and reward maximization. We use a Bimodal Bandit problem, in Figure 1 (Left), with a Gaussian base policy and GM policy with two components. We obtain stationary points $\{\boldsymbol{\theta} \mid \nabla_{\boldsymbol{\theta}} J(\pi_{\boldsymbol{\theta}}) = \mathbf{0}\}$ for both by running a local gradient descent optimization on (8) with various $\alpha$ from 100 different random starting points. More details are in Appendix C.

**Robustness to entropy regularization.** Figure 1 (Middle) shows the expected reward $J_0(\pi_{\boldsymbol{\theta}^*})$ of the optimal stationary points for the two policies. The points with zero expected reward indicate no stationary point found. The GM dominates for $\alpha \in (0.2, 0.45]$. More specifically, for $\alpha \in (0.2, 0.3]$, the optimal Gaussian policy cannot concentrate its mode in any of the reward mode while the GM policy can concentrate on both (see Figure 1 (Left) for $\alpha = 0.3$). For $\alpha \in (0.3, 0.45]$, the Gaussian policy does not find any stationary points, while the GM policy continues to perform reasonably.

**Preference for multimodality increases with larger entropy regularization.** The GM policy can be either unimodal and bimodal. We plot the ratio between these two types found in the 100 trials in Figure 1 (Right). We can see a general trend: as $\alpha$ increases, the frequency of finding bimodal policies also increases. This effect is most pronounced after $\alpha = 0.3$, where all policies found are bimodal. This result suggests that a high entropy scale can help prevent mode collapse in mixture policies, a phenomenon where the policy loses its multimodality and becomes effectively unimodal.

## 4 REPARAMETERIZED GRADIENT ESTIMATORS FOR MIXTURE POLICIES

In this section, we derive reparameterization gradient estimators for mixture policies. The mixture policy is hard to reparameterize because of the softmax weighting policy, even though the individual Gaussian component policies are easy to reparameterize. We provide two approaches: a partial reparameterization, where we only reparameterize component policies, giving an unbiased gradient, and a full reparameterization using a Gumbel-softmax, which results in a biased reparameterization.

## 4.1 Reparameterization of Component Policies

We extend the reparameterization policy gradient theorem (Lan et al., 2022) to the case of mixture policies where we reparameterize only the components. This half-reparameterization remains unbiased, though we cannot be certain that it has the same variance reduction properties as a typical reparameterization. Intuitively, it should reduce some variance due to the component policies, and we do find it has empirical benefits later. We assume the component policies are reparameterized as $\pi_{\boldsymbol{\theta}}^b(a|s,k) = p(\epsilon)$, where $a = f_{\boldsymbol{\theta}}(\epsilon; s, k)$ and $p(\cdot)$ is the corresponding noise distribution. Appendix B contains the proofs, and the result for the unregularized setting is obtained by setting $\alpha = 0$.

**Assumption 4.1.** $\mathcal{S}$ and $\mathcal{A}$ are compact.

**Assumption 4.2.** $p(s'|s,a)$, $d_0(s)$, $r(s,a)$ $f_{\boldsymbol{\theta}}(\epsilon; s, k)$, $f_{\boldsymbol{\theta}}^{-1}(a; s, k)$, $\pi_{\boldsymbol{\theta}}^w(k|s)$, $\pi_{\boldsymbol{\theta}}^b(a|s,k)$, $p(\epsilon)$, and their derivatives are continuous in variables $s$, $a$, $s'$, $\boldsymbol{\theta}$, and $\epsilon$.

**Theorem 4.3** (Entropy-Regularized Half-Reparameterization Policy Gradient Theorem). *Under Assumptions 4.1 and 4.2, we have*

$$\nabla_{\boldsymbol{\theta}} J(\pi_{\boldsymbol{\theta}}^m) = \mathbb{E}_{s \sim d_{\pi_{\boldsymbol{\theta}}^m}, k \sim \pi_{\boldsymbol{\theta}}^w(\cdot|s), \epsilon \sim p} \Big[ \nabla_{\boldsymbol{\theta}} \log \pi_{\boldsymbol{\theta}}^w(k|s) \big( Q_{\pi_{\boldsymbol{\theta}}^m}(s, f_{\boldsymbol{\theta}}(\epsilon; s, k)) - \alpha \log \pi_{\boldsymbol{\theta}}^m(f_{\boldsymbol{\theta}}(\epsilon; s, k)|s) \big)$$
$$+ \nabla_{\boldsymbol{\theta}} f_{\boldsymbol{\theta}}(\epsilon; s, k) \nabla_a \big( Q_{\pi_{\boldsymbol{\theta}}^m}(s, a) - \alpha \log \pi_{\boldsymbol{\theta}}^m(a|s) \big)|_{a = f_{\boldsymbol{\theta}}(\epsilon; s, k)} \Big],$$

*where $d_{\pi_{\boldsymbol{\theta}}^m}(s) \doteq \sum_{t=0}^{\infty} \mathbb{E}_{\pi_{\boldsymbol{\theta}}^m}[\gamma^t \mathbb{I}(S_t = s)]$ is the (discounted) occupancy measure under $\pi_{\boldsymbol{\theta}}^m$.*

Similarly, we can obtain the half-reparameterization gradient of the objective of SAC in (3).

**Assumption 4.4.** $Q_{\mathbf{w}}(s,a)$ and its derivatives are continuous in variables $s$ and $a$.

**Proposition 4.5.** *Under Assumptions 4.1, 4.2, and 4.4, we have*

$$\nabla_{\boldsymbol{\theta}} \hat{J}(\pi_{\boldsymbol{\theta}}^m) = \mathbb{E}_{s \sim \mathcal{B}, k \sim \pi_{\boldsymbol{\theta}}^w(\cdot|s), \epsilon \sim p} \Big[ \nabla_{\boldsymbol{\theta}} \log \pi_{\boldsymbol{\theta}}^w(k|s) \big( Q_{\mathbf{w}}(s, f_{\boldsymbol{\theta}}(\epsilon; s, k)) - \alpha \log \pi_{\boldsymbol{\theta}}^m(f_{\boldsymbol{\theta}}(\epsilon; s, k)|s) \big)$$
$$+ \nabla_{\boldsymbol{\theta}} \big( Q_{\mathbf{w}}(s, f_{\boldsymbol{\theta}}(\epsilon; s, k)) - \alpha \log \pi_{\boldsymbol{\theta}}^m(f_{\boldsymbol{\theta}}(\epsilon; s, k)|s) \big) \Big].$$

From Proposition 4.5, we can obtain the *half-reparameterization* estimator for SAC's objective:

$$\hat{\nabla}_{\boldsymbol{\theta}} \hat{J}(\pi_{\boldsymbol{\theta}}^m) = \nabla_{\boldsymbol{\theta}} \log \pi_{\boldsymbol{\theta}}^w(K_t|S_t) \big( Q_{\mathbf{w}}(S_t, f_{\boldsymbol{\theta}}(\epsilon_t; S_t, K_t)) - \alpha \log \pi_{\boldsymbol{\theta}}^m(f_{\boldsymbol{\theta}}(\epsilon_t; S_t, K_t)|S_t) \big)$$
$$- \nabla_{\boldsymbol{\theta}} \big( Q_{\mathbf{w}}(S_t, f_{\boldsymbol{\theta}}(\epsilon_t; S_t, K_t)) - \alpha \log \pi_{\boldsymbol{\theta}}^m(f_{\boldsymbol{\theta}}(\epsilon_t; S_t, K_t)|S_t) \big). \tag{9}$$

## 4.2 Reparameterization of Weighting Policies with Gumbel-Softmax

Since the output of the weighting policy $\pi_{\boldsymbol{\theta}}^w(\cdot|s)$ for any state $s$ is a categorical distribution, we can not directly reparameterize it. However, there are various approaches to obtain biased reparameterized samples from it (Bengio et al., 2013; Maddison et al., 2016; Jang et al., 2016). Here, we use the straight-through Gumbel-softmax reparameterization (Jang et al., 2016).

Given a weighting distribution $\pi_{\boldsymbol{\theta}}^w(\cdot|s)$ and i.i.d samples from Gumbel$(0,1)$, $g_1, \cdots, g_N$, we can obtain a sample from the corresponding Gumbel-softmax distribution:

$$y_{\boldsymbol{\theta}}(\mathbf{g}; s, k) = \frac{\exp\left(\log \pi_{\boldsymbol{\theta}}^w(k|s) + g_k\right)/\tau}{\sum_{k'=1}^{N} \exp\left(\log \pi_{\boldsymbol{\theta}}^w(k'|s) + g_{k'}\right)/\tau} \quad \text{for } k = 1, \ldots, N,$$

where we define $\mathbf{g} = [g_1, \cdots, g_N]$. Using the Gumbel-max trick, we can obtain a sample from $\pi_{\boldsymbol{\theta}}^w(\cdot|s)$ using the Gumbel-softmax sample $y_{\boldsymbol{\theta}}(\mathbf{g}; s, k)$:

$$\hat{\mathbf{z}} = \text{one\_hot}\left( \arg\max_k \left( y_{\boldsymbol{\theta}}(\mathbf{g}; s, k) \right) \right) = \text{one\_hot}\left( \arg\max_k \left( \log \pi_{\boldsymbol{\theta}}^w(k|s) + g_k \right) \right).$$

We can further use the straight-through trick to obtain a differentiable one-hot sample $\mathbf{z} = [z_{\boldsymbol{\theta}}(\mathbf{g}; s, 1), \cdots, z_{\boldsymbol{\theta}}(\mathbf{g}; s, N)]$, where $z_{\boldsymbol{\theta}}(\mathbf{g}; s, k)$ is defined as follows:

$$z_{\boldsymbol{\theta}}(\mathbf{g}; s, k) = \hat{z}_k + y_{\boldsymbol{\theta}}(\mathbf{g}; s, k) - y_{\boldsymbol{\phi}}(\mathbf{g}; s, k)|_{\boldsymbol{\phi} = \boldsymbol{\theta}} \quad \text{for } k = 1, \ldots, N.$$

Finally, using the differential one-hot sample $\mathbf{z}$ from the weighting policy $\pi_{\boldsymbol{\theta}}^w$ and reparameterized samples from the component policies, we can obtain a differentiable action sample $a$:

$$a = \sum_{k=1}^{N} z_{\boldsymbol{\theta}}(\mathbf{g}; s, k) f_{\boldsymbol{\theta}}(\epsilon; s, k). \tag{10}$$

Plugging (10) back to (5), we can obtain a full reparameterization estimator, which we call the *Gumbel-reparameterization* estimator:

$$\hat{\nabla}_{\boldsymbol{\theta}} \hat{J}(\pi_{\boldsymbol{\theta}}^m) = \nabla_{\boldsymbol{\theta}} \big( Q_{\mathbf{w}}(S_t, A_t) - \alpha \log \pi_{\boldsymbol{\theta}}^m(A_t | S_t) \big) \text{ where } A_t = \sum_{k=1}^N z_{\boldsymbol{\theta}}(\mathbf{g}_t; S_t, k) f_{\boldsymbol{\theta}}(\epsilon_t; S_t, k) \quad (11)$$

**The temperature parameter $\tau$ controls a bias-variance trade-off.** When $\tau$ approaches 0, the soft sample $\mathbf{y} = [y_{\boldsymbol{\theta}}(\mathbf{g}; s, 1), \cdots, y_{\boldsymbol{\theta}}(\mathbf{g}; s, N)]$ will converge to a one-hot vector and recover the categorical sample. However, the variance of the gradients with respect to $\pi_{\boldsymbol{\theta}}^w(\cdot | s)$ will increase. On the other hand, when $\tau$ becomes larger, the variance of the gradients will decrease, but the soft sample $\mathbf{y}$ will converge to a uniform vector. See Jang et al. (2016) for detailed discussions. In our study, we find that using a fixed temperature $\tau = 1$ works well.

## 5 EXPERIMENTS

In this section, we conduct experiments to investigate the utility of mixture policies in continuous control problems. We use SAC as the base learning algorithm and the squashed Gaussian policy as the base policy. We denote different SAC instances using X-Y, where X represents the policy's gradient estimator, and Y represents the policy's parameterization. For the base squashed Gaussian policy , we consider the likelihood (Like-Squashed) and the reparameterization (Rep-Squashed) estimators. For the squashed Gaussian mixture (SGM) policy, we consider the likelihood (Like-SGM), the half-reparameterization (HalfRep-SGM), and the Gumbel-reparameterization (GumbelRep-SGM) estimators. We use a mixture of 5 components for the mixture policy in all our experiments, while we also provide a study on the effect of the number of components in Appendix E.6.

### 5.1 SOLVING CONTINUOUS CONTROL PROBLEMS WITH MIXTURE POLICIES

We first investigate the performance of mixture policies in 19 environments from two commonly used continuous control benchmarks: 7 MuJoCo environments from OpenAI Gym (Brockman et al., 2016) and 12 environments from the DeepMind Control Suite (Tassa et al., 2018). We use SAC with automatic entropy tuning and the hyperparameters reported in the SAC paper (Haarnoja et al., 2018c), which are tuned based on Rep-Squashed. The likelihood-ratio estimator is relatively less stable in these high-dimensional environments, so we will focus on comparing Rep-Squashed with HalfRep-SGM and GumbelRep-SGM while noting that the used hyperparameters may not be the best for the mixture policies. We use 8 random seeds in these experiments. Please refer to Appendix D for more implementation details.

**Mixture policies aren't helpful when the rewards are shaped.** Figure 2 shows the learning curves in twelve selected environments. We can see that the performances of the mixture variants are quite similar to the base policy across different environments except in cartpole: swingup_sparse, where they have better average performance though with overlapping confidence intervals. Among these environments, only cartpole: balance_sparse, ball_in_cpu: catch, and cartpole: swingup_sparse have sparse rewards, while other environments have shaped rewards to guide policy learning. Thus, it is not surprising that the mixture policy is not helping here. While cartpole: balance_sparse and ball_in_cpu: catch are too easy for all policies, it is interesting to see that the mixture policy has a better performance in cartpole: swingup_sparse. The observation in the remaining seven environments is consistent with our discussion here, please refer to Appendix D for more details.

### 5.2 MIXTURE POLICIES IMPROVE LEARNING WHEN THE REWARDS ARE UNSHAPED

Based on the results in Figure 2, we hypothesize that *mixture policies improve exploration in environments with unshaped rewards but not in those with shaped rewards*. In this section, we design experiments and analysis to verify this hypothesis.

To perform a more extensive empirical investigation with proper hyperparameter tuning, we use three classic control environments in this section: Pendulum, Acrobot, and MountainCar. Specifically, we use two different variants for each of them: one with shaped rewards (ShapedPendulum, ShapedAcrobot, and ShapedMountainCar) and the other one with unshaped rewards (Pendulum, Acrobot, and MountainCar). We refer the reader to Appendix D about their specific reward functions. In this experiment, we use SAC with a fixed entropy scale as it is reported that SAC with

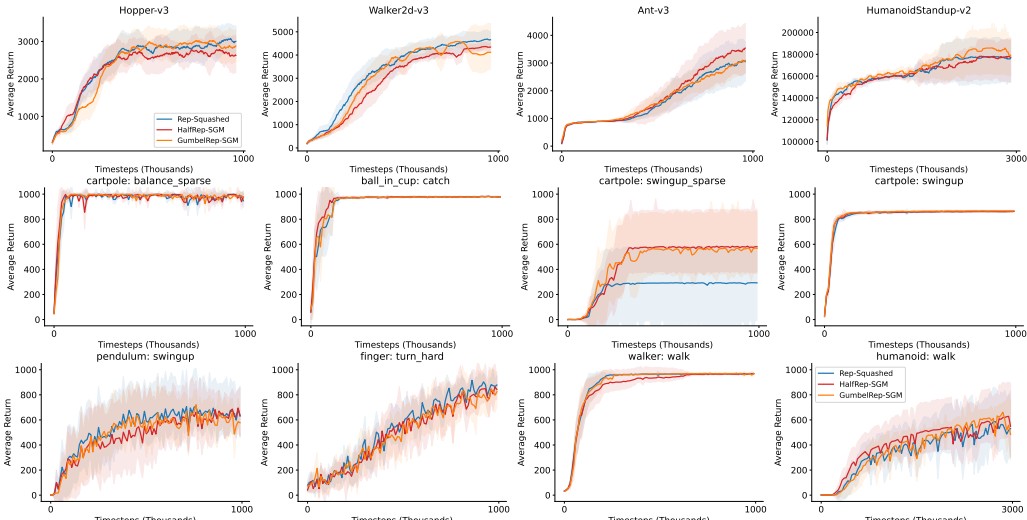

Figure 2: Learning curves in four MuJoCo and eight DeepMind Control Suite environments. The shaded area shows the $95\%$ bootstrap confidence intervals across 8 runs.

automatic entropy performs worse in this domain (Neumann et al., 2022). Specifically, we sweep the entropy scale $\alpha = 10^y$ for $y \in \{-3, -2, -1, 0\}$. In addition, we sweep the initial critic step size $\eta_{q,0} = 10^x$ for $x \in \{-5, -4, -3, -2\}$ and the initial actor step size $\eta_{p,0} = \kappa\eta_{q,0}$ for $\kappa \in \{10^{-2}, 10^{-1}, 1, 10\}$. We run each hyperparameter setting for 10 runs and report another 30 reruns of the best setting that has the largest area under the learning curve (AUC).

**Mixture policies improve exploration when the rewards are shaped.** The top two rows of Figure 4 show the learning curves of the best hyperparameter setting. By comparing the two rows, we can see that the performances in the environments with shaped rewards are much stabler than those with unshaped rewards. While the mixture policy is performing quite similarly to the base policy in the former case, the learning curve of the mixture policy is consistently above the corresponding variant of the base policy in the latter. Note that it is reasonable to see such large performance variation in these environments because the agent is only rewarded when reaching goal states and receives very few reward signals otherwise. Figure 3 plots the individual runs for Rep-Squashed and HalfRep-SGM in Acrobot, where we can intuitively see the failing seeds will induce wide confidence intervals. In addition, we can also see that HalfRep-SGM has more seeds that find the goal states after the initial training phase, suggesting its more persistent exploration.

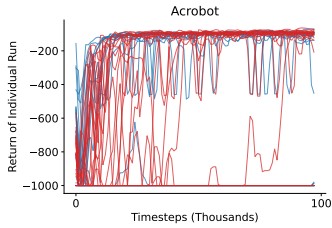

Figure 3: Learning curves for individual runs of Rep-Squashed (blue) and HalfRep-SGM (red) in Acrobot.

**Mixture policies are more robust to the entropy regularization.** The bottom row of Figure 4 shows the sensitivity to the entropy scale for Rep-Squashed, HalfRep-SGM, and GumbelRep-SGM in environments with unshaped rewards. Despite the bootstrap confidence intervals being again quite large, we can see that the sensitivity curves of HalfRep-SGM and GumbelRep-SGM are mostly above Rep-Squashed, suggesting the mixture policy is relatively robust to the entropy scale compared to the base policy.

**The critic is less smooth when the rewards are unshaped.** To shed some light on why mixture policies may behave more differently from the base policy when the rewards are unshaped, we plot the learning curve, action-value estimates, and the policy density at a starting state close to the bottom in both ShapedMountainCar and MountainCar. From Figure 5, we can see that the mixture policy maintains its multimodality more often in the MountainCar, where the rewards are unshaped. Note that the agent at the starting state needs to decide which way to start to accumulate

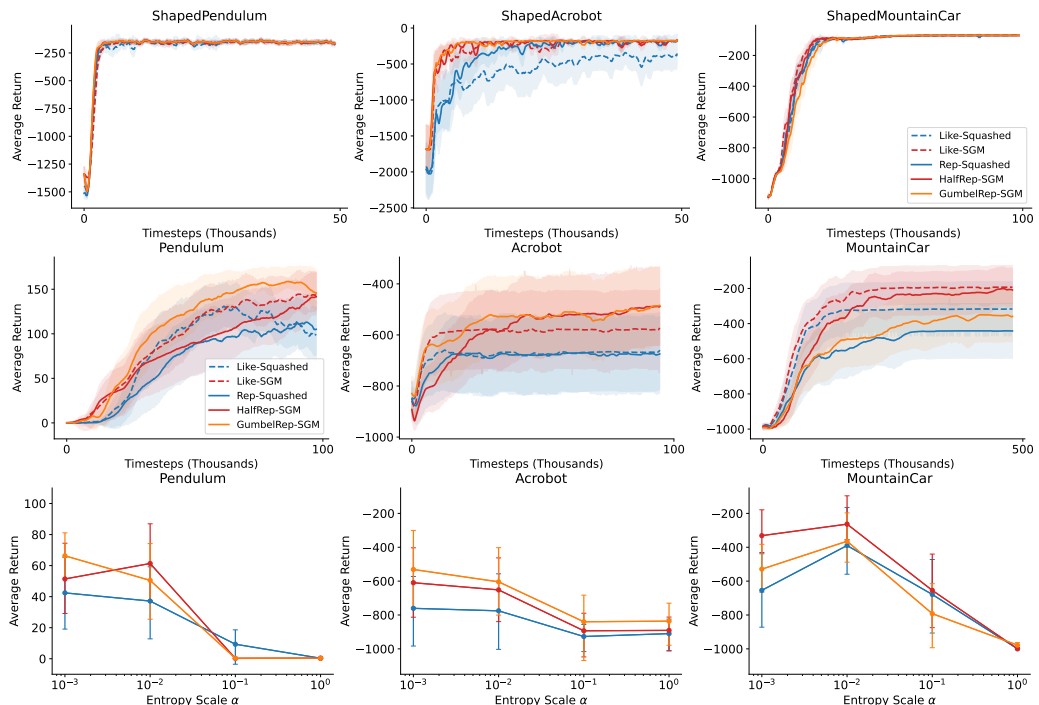

Figure 4: Learning curves and sensitivity curves for classic control environments. The shaded area shows the 95% bootstrap confidence intervals across 30 runs. The error bars plot the 95% bootstrap confidence intervals across 10 runs.

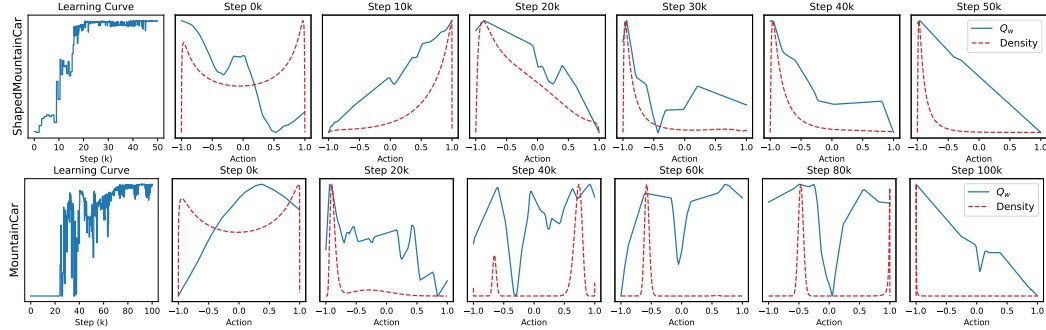

Figure 5: Learning curve, action-value estimates, and policy density at a starting state from a sample run of HalfRep-SGM in two MountainCar variants. The y-axes differ across plots and are not shown with ticks to highlight the shape of the curves rather than their exact values. The learning curves show faster convergence and smoother performance in ShapedMountainCar with shaped rewards, whereas unshaped rewards in MountainCar lead to more erratic and slower learning. The action-value estimates are smoother and more stable in ShapedMountainCar. In contrast, they are more unstable and multimodal in MountainCar. Correspondingly, the density quickly becomes unimodal and concentrates on one of the boundaries in ShapedMountainCar, while multimodal density has more occurrence in MountainCar, reflecting continued exploration of the mixture policy.

its momentum to explore states farther away. In this case, maintaining a multimodal density may help the agent explore hills on both sides. The multimodality of the policy may explain why the mixture policy explores better in this case. On the other hand, in ShapedMountainCar, the critic and the actor (policy) are guided by the shaped rewards, resulting in much smoother action-value estimates with fewer modes and reducing the need to explore multiple directions. This may also be why mixture policies don't provide many benefits in terms of exploration in those environments with shaped rewards presented in Figure 2 because there is no or minimum need to explore.

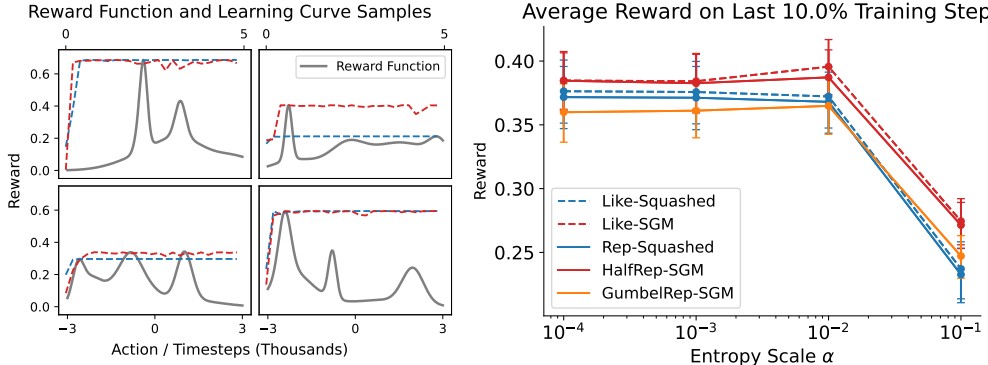

Figure 6: Left: Examples of randomly generated multimodal bandits and the corresponding learning curves on them. Right: Performance across different $\alpha$. Each dot represents the final performance of the best hyperparameter setting under the corresponding $\alpha$. The error bars show the $95\%$ bootstrap confidence intervals across $100$ different bandits. We can see that SGM with unbiased estimators (Like and HalfRep) outperforms squashed Gaussian consistently. In addition, the gap between their performance increases as $\alpha$ increases.

## 5.3 DISENTANGLING FROM THE NONSTATIONARITY OF THE CRITIC

We have seen in the last section that the critic can sometimes be multimodal, and mixture policies can explore multiple rewarding directions at the same time with a multimodal density in this case. In this section, we demonstrate their another benefit that is not clearly shown: *Mixture policies can more easily identify higher peaks in the critic's value function compared to the base policy.* This is not obvious if we examine the critic and the actor during the learning process in an MDP as in Figure 5, as the critic is constantly changing. Thus, to understand this effect, we investigate it in the simpler bandit setting, where the critic (i.e., the reward function) is given to the agent and stationary.

Specifically, we randomly generate 100 continuous bandits with the reward function proportional to the summation of 30 Gaussian density functions. The means and standard deviations are uniformly sampled from $[-3, 3]$ and $[0.1, 1.0]$, respectively. Figure 6 (Left) shows a few examples of the generated bandits and the learning curves corresponding to them. We sweep the initial actor step size $\eta_{p,0} = 10^x$ for $x \in \{-4, -3, -2, -1\}$ and the entropy scale $\alpha = 10^y$ for $y \in \{-4, -3, -2, -1\}$. We run each hyperparameter setting for one run on each bandit. We report results of the best setting that has the highest average reward over the last $10.0\%$ training steps, as we are mostly interested in how well the agent explores. We also report results based on AUC and other experimental details in Appendix D.

**Mixture policies explore more efficiently.** Figure 6 (Right) shows the aggregated final performance of all five algorithm instances. We can see that the SGM policy is better than the squashed Gaussian policy when using either the likelihood-ratio gradient estimator or the unbiased half-reparameterization gradient estimator across different $\alpha$. On the other hand, the SGM policy with the Gumbel-reparameterization gradient estimator is worse than the counterpart of the base policy. We hypothesize that the reason is that this gradient estimator is biased and will learn slower in simple environments like this, which allows very large step sizes.

**Mixture policies also improve robustness.** Another observation from Figure 6 (Right) is that mixture policies are more robust to the entropy scale, which is consistent with our results presented in Section 3. Specifically, other than the consistent improvement of the SGM policy with unbiased estimators over the base policy, we can see that the gap between them increase as $\alpha$ increases. In addition, the gap between the base policy and the SGM policy with a biased gradient (GumbelRep-SGM) appears to shrink as $\alpha$ increases. This result suggests that mixture policies are possibly more robust to larger entropy scales and explore more efficiently with a moderate large entropy scale. This is not the case for the base policy in this experiment.

## 6 CONCLUSIONS

Mixture policies are a simple way to increase the flexibility of the policy parameterization, but very little has been documented about their efficacy, or lack of efficacy. Our aim was to start to fill this gap, to make this a more accessible tool when using entropy-regularized actor-critic algorithms with continuous actions, like Soft-Actor Critic. The clear outcome from the study is that mixture policies are comparable, and sometimes notably better than, a base unimodal policy. Through a few basic theoretical results and experiments in bandits, we highlighted that mixture policies are more robust to entropy scale, with 1) a preference for multimodality increasing with higher entropy, 2) divergence (lack of stationary points) for the base Gaussian policy, unlike the mixture, 3) better balance between entropy and the expected reward objective, resulting in higher unregularized values in addition to higher regularized values and 4) higher likelihood of finding maxima on a multimodal surface. This behavior seemed to manifest in better exploration in environments with unshaped, or uninformative rewards; in such environments, without shaped rewards, exploration is critical and the mixture policies performed better than the unimodal base policy. In particular, we found the base policy had more failed runs where it was unable to find the goal at all.

To leverage the utility of mixture policies, however, we needed a small algorithmic improvement: a reparameterization gradient. We proposed two new reparameterization gradient estimators for mixture policies, filling in a gap in the literature. The first estimator is a half-reparameterization, only reparameterizing the component policies and not the softmax weighting policy. This estimator provides an unbiased estimate but does not fully reparameterize, potentially not obtaining the same variance reduction properties. We provided a full reparameterization by using the Gumbel-reparameterization for the softmax, giving up unbiasedness. Despite this bias, this estimator was still effective, though we found it to be more sensitive than the half-reparameterization gradient.

There are several limitations of this work. The primary limitation is that we used default hyperparameters for the mixture policies in MuJoCo; such defaults were tuned for SAC with the base policy. Hyperparameter tuning in large domains is extremely expensive, and arguably overtuning can also produce misleading results. Characterizing if there are differences in effective hyperparameters for these different policy parameterization, however, would provide a more complete picture. Otherwise, as with any empirical study, there is always a limitation in scope, though arguably this limitation is necessary to make progress. We dove deeply into SAC, and started with an exploratory study across standard benchmarks. The study highlighted that we should expect to see bigger differences in environments without shaped rewards, and sets up a clear question for a follow-up study.

### REPRODUCIBILITY STATEMENT

We present proofs for all theoretical results in Appendix B. We also enclose relevant details about all empirical investigations with our best effort in Appendices C, D, and F. In addition, we provide additional experiments in Appendix E to support the generality of our findings. We will release the source code for all our experiments before publication.

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

## A RELATED WORKS

In this section, we provide a more in-depth discussion of related work, beyond that given in the introduction.

**Implicit policies.** Existing works on modeling continuous distributions on continuous action spaces can be put into two categories: Policies using parametric distributions and implicit policies. We have discussed the former in the introduction of our paper. Here, we discuss the latter. Implicit policies utilize deep generative models (e.g., energy-based models; Haarnoja et al., 2017; Messaoud et al., 2024; normalizing flows; Tang & Agrawal, 2018; Mazoure et al., 2020; diffusion models; Wang et al., 2023) to model the policy. These models can model complex distributions and have improved learning efficiency, but they usually have more parameters and complex training pipelines. Compared to these more complex policies, policies using parametric distributions have several benefits. Firstly, they are simpler and more efficient to train. Secondly, they have simple explicit probability density, which is useful in various ways in entropy-regularized RL. Note that some implicit policies don't hold such a property. Further, mixture policies, which we consider in our paper, also have modeling power to model arbitrary distribution given a sufficiently large number of components. Given these benefits, we think it is important to improve our understanding of mixture policies. Nevertheless, we agree that it is also important to investigate more complex but powerful implicit policies, and the benefits of mixture policies demonstrated in our paper can potentially be generalized to them.

**Other uses of mixture policies.** Since mixture distributions are a widely known model, mixture policies have been investigated in various ways in the literature. Daniel et al. (2012) and Celik et al. (2022) focus on exploiting the hierarchy in mixture policies for problems with hierarchical structures. In their work, they design algorithms specific to mixture policies. Sharing the same motivation to use mixture policies to model diverse behaviors, Nematollahi et al. (2022) learn a prior GMM using imitation learning and then use SAC to learn the changes to the prior GMM for adaptation. Here, the action space of SAC is the changes to GMM's parameters. With a different motivation, Seyde et al. (2022) use mixture policies to select from a diverse set of sub-policies to reduce the hyperparameter sensitivity of the algorithm.

**Mixture policies as the policy parameterizations for SAC.** Different from these previous works, our motivation for using mixture policies is to treat them as a more complex policy class and understand the effect of a more complex policy class under the entropy-regularization setting. Thus, our treatment does not include designing specific objective functions for mixture policies but using the standard regularized objective that is agnostic to policy parameterizations. In this regard, the closest related works are Haarnoja et al. (2018b), Hou et al. (2020), and Baram et al. (2021). In the first version of SAC, Haarnoja et al. did indeed test mixture policies but then did not pursue this further nor provide insights on this choice. We hypothesize the lack of reparameterization estimators for mixture policies might be the reason that later versions of SAC switched to a single Gaussian as they found the reparameterization gradient estimator works better (see Footnote 3 on Page 67 of Haarnoja, 2018). Later, Hou et al. try to avoid reparameterization of the whole mixture policy in SAC by using a separate objective for the weighting policy. However, their evaluation on a restrictive set of MuJoCo environments doesn't show a significant improvement from their approach. Further, Baram et al. also revisit SAC with a mixture policy. However, their focus is to explore the utility of the upper and lower bounds of mixture models' entropy, without considering a learnable weighting policy.

It is not highly novel to use mixture policies, but rather to understand the effect of doing so. In our work: 1) We provide new insights into the effect of a more flexible policy class in the stationary points of the entropy regularized objective; 2) we are the first to propose and study reparameterization gradient estimators for mixture policies, filling in a gap in the literature; and 3) we explore the benefits of and provide insights into using mixture policies in environments with unshaped rewards, complementing existing works on using mixture policies in entropy-regularized actor-critic.

# B PROOFS

## B.1 PROOFS FOR RESULTS IN SECTION 3

**Proposition 3.1.** When both $\pi^b_{\boldsymbol{\theta}^{b,*}}$ and $\pi^m_{\boldsymbol{\theta}^{m,*}}$ exist, then $J(\pi^m_{\boldsymbol{\theta}^{m,*}}) \geq J(\pi^b_{\boldsymbol{\theta}^{b,*}})$.

*Proof.* Since the policy class of $\pi^m_{\boldsymbol{\theta}^m}$ is a super set of the policy class of $\pi^b_{\boldsymbol{\theta}^b}$, it is apparent that the optimal value of the policy class of $\pi^m_{\boldsymbol{\theta}^m}$ is better than that of $\pi^b_{\boldsymbol{\theta}^b}$. $\qquad\square$

**Proposition 3.2.** Consider entropy-constrained policy optimization $\max_{\boldsymbol{\theta}} J_0(\pi_{\boldsymbol{\theta}})$ subject to $\mathcal{H}(\pi_{\boldsymbol{\theta}}) \geq H$ for some $H > 0$ and define the optimal solution $\boldsymbol{\theta}'$, then $J_0(\pi^m_{\boldsymbol{\theta}^{m,\prime}}) \geq J_0(\pi^b_{\boldsymbol{\theta}^{b,\prime}})$.

*Proof.* Define $\tilde{\boldsymbol{\theta}}^m = [\boldsymbol{\theta}^{b,\prime\top}, \cdots, \boldsymbol{\theta}^{b,\prime\top}, \boldsymbol{\theta}^{w\top}]^\top$ for arbitrary $\boldsymbol{\theta}^w$. Apparently, $\pi^m_{\tilde{\boldsymbol{\theta}}^m}(a) = \pi^b_{\boldsymbol{\theta}^{b,\prime}}(a)$. Then,

$$J_0(\pi^m_{\boldsymbol{\theta}^{m,\prime}}) \geq J_0(\pi^m_{\tilde{\boldsymbol{\theta}}^m}) = J_0(\pi^b_{\boldsymbol{\theta}^{b,\prime}}).$$

$\qquad\square$

**Proposition 3.3.** Assume $r : \mathcal{A} \to \mathbb{R}$ is an integrable function on $\mathcal{A} = \mathbb{R}$. For all $\alpha > \frac{3}{2} r_{\max}$, $J(\pi_{\mu,\sigma}) = \mathbb{E}_{a \sim \mathcal{N}(\mu,\sigma)}[r(a) - \alpha \log \mathcal{N}(a; \mu, \sigma)]$ doesn't have any stationary point.

*Proof.* To show that $J(\pi_{\mu,\sigma})$ doesn't have any stationary point, it is sufficient to show that its partial derivative with respect to $\sigma$ is lower bounded by zero:

$$\frac{\partial J(\pi_{\mu,\sigma})}{\partial \sigma} > 0, \quad \forall \mu \in \mathbb{R}, \sigma > 0. \tag{12}$$

We first simplify the entropy term $H(\mathcal{N}(\cdot; \mu, \sigma))$ for the Gaussian policy:

$$
\begin{aligned}
H(\mathcal{N}(\cdot; \mu, \sigma)) &= -\mathbb{E}_{a \sim \mathcal{N}(\mu,\sigma)}[\log \mathcal{N}(a; \mu, \sigma)] \\
&= -\mathbb{E}_{a \sim \mathcal{N}(\mu,\sigma)} \left[ \log \left( \frac{1}{\sqrt{2\pi\sigma^2}} \exp \left( -\frac{(a-\mu)^2}{2\sigma^2} \right) \right) \right] \\
&= \frac{1}{2} \log(2\pi\sigma^2) + \frac{1}{2\sigma^2} \mathbb{E}_{a \sim \mathcal{N}(\mu,\sigma)}[(a-\mu)^2] \\
&= \frac{1}{2} \log(2\pi\sigma^2) + \frac{1}{2}.
\end{aligned}
$$

Further, we can derive its partial derivative with respect to $\sigma$: $\frac{\partial H(\mathcal{N}(\cdot; \mu, \sigma))}{\partial \sigma} = \frac{1}{\sigma}$.

Define $r(\mu, \sigma) = \mathbb{E}_{a \sim \mathcal{N}(\mu,\sigma)}[r(a)] = \int_a r(a) \mathcal{N}(a|\mu, \sigma)\, da$. Then,

$$
\begin{aligned}
\frac{\partial J(\pi_{\mu,\sigma})}{\partial \sigma} &= \frac{\partial}{\partial \sigma} \mathbb{E}_{a \sim \mathcal{N}(\mu,\sigma)}[r(a) - \alpha \log \mathcal{N}(a; \mu, \sigma)] \\
&= \frac{\partial}{\partial \sigma} \left( r(\mu, \sigma) + \alpha H(\mathcal{N}(\cdot; \mu, \sigma)) \right) \\
&= \frac{\partial r(\mu, \sigma)}{\partial \sigma} + \frac{\alpha}{\sigma}.
\end{aligned}
$$

To show (12), we just need to show

$$\sigma \frac{\partial r(\mu, \sigma)}{\partial \sigma} > -\alpha. \tag{13}$$

We first analyze the left hand side of (13):

$$\sigma \frac{\partial r(\mu, \sigma)}{\partial \sigma} = \sigma \frac{\partial}{\partial \sigma} \int_a r(a) \mathcal{N}(a|\mu, \sigma) \, da$$

$$= \sigma \int_a r(a) \frac{\partial}{\partial \sigma} \mathcal{N}(a|\mu, \sigma) \, da$$

$$= \sigma \int_a r(a) \frac{\partial}{\partial \sigma} \left( \frac{1}{\sqrt{2\pi\sigma^2}} \exp\left(-\frac{1}{2\sigma^2}(a-\mu)^2\right) \right) \, da$$

$$= \sigma \int_a r(a) \left( -\frac{1}{\sqrt{2\pi\sigma^4}} + \frac{(a-\mu)^2}{\sqrt{2\pi\sigma^8}} \right) \exp\left(-\frac{1}{2\sigma^2}(a-\mu)^2\right) \, da$$

$$= \int_a r(a) \frac{1}{\sqrt{2\pi\sigma^2}} \exp\left(-\frac{(a-\mu)^2}{2\sigma^2}\right) \left( \frac{(a-\mu)^2}{\sigma^2} - 1 \right) \, da$$

$$\overset{b=\frac{a-\mu}{\sigma}}{=} \int_b r(\sigma b + \mu) \frac{1}{\sqrt{2\pi}} \exp\left(-\frac{b^2}{2}\right) \left( \frac{b^2}{2} - 1 \right) \, db,$$

which is bounded:

$$\left| \sigma \frac{\partial r(\mu, \sigma)}{\partial \sigma} \right| = \left| \int_b r(\sigma b + \mu) \frac{1}{\sqrt{2\pi}} \exp\left(-\frac{b^2}{2}\right) \left( \frac{b^2}{2} - 1 \right) \, db \right|$$

$$\leq \int_b |r(\sigma b + \mu)| \frac{1}{\sqrt{2\pi}} \exp\left(-\frac{b^2}{2}\right) \left| \frac{b^2}{2} - 1 \right| \, db$$

$$\leq \int_b r_{\max} \frac{1}{\sqrt{2\pi}} \exp\left(-\frac{b^2}{2}\right) \left| \frac{b^2}{2} - 1 \right| \, db$$

$$\leq r_{\max} \int_b \frac{1}{\sqrt{2\pi}} \exp\left(-\frac{b^2}{2}\right) \left( \frac{b^2}{2} + 1 \right) \, db$$

$$\leq r_{\max} \mathbb{E}_{b \sim \mathcal{N}(0,1)} \left[ \frac{b^2}{2} + 1 \right]$$

$$= \frac{3}{2} r_{\max}.$$

Then for any $\alpha > \frac{3}{2} r_{\max}$, we have $\sigma \frac{\partial r(\mu,\sigma)}{\partial \sigma} \geq -\frac{3}{2} r_{\max} > -\alpha$. $\qquad\square$

**Proposition 3.4.** For arbitrary $\boldsymbol{\theta}^w$ and any $\tilde{\boldsymbol{\theta}}^b$ such that $\nabla_{\boldsymbol{\theta}^b} J(\pi_{\tilde{\boldsymbol{\theta}}^b}^b) = 0$, we have $\nabla_{\boldsymbol{\theta}^m} J(\pi_{\tilde{\boldsymbol{\theta}}^m}^m) = 0$, where $\tilde{\boldsymbol{\theta}}^m = \left[ \tilde{\boldsymbol{\theta}}^{b\top}, \cdots, \tilde{\boldsymbol{\theta}}^{b\top}, \boldsymbol{\theta}^{w\top} \right]^\top$.

*Proof.* By assumption,

$$\nabla_{\boldsymbol{\theta}^b} J(\pi_{\tilde{\boldsymbol{\theta}}^b}^b) = \nabla_{\boldsymbol{\theta}^b} \mathbb{E}_{a \sim \pi_{\tilde{\boldsymbol{\theta}}^b}^b(a)} \left[ r(a) - \alpha \log \pi_{\tilde{\boldsymbol{\theta}}^b}^b(a) \right]$$

$$= \nabla_{\boldsymbol{\theta}^b} \int_a \pi_{\tilde{\boldsymbol{\theta}}^b}^b(a) \left( r(a) - \alpha \log \pi_{\tilde{\boldsymbol{\theta}}^b}^b(a) \right) \, da$$

$$= \int_a \left( r(a) - \alpha \log \pi_{\tilde{\boldsymbol{\theta}}^b}^b(a) - \alpha \right) \nabla_{\boldsymbol{\theta}^b} \pi_{\tilde{\boldsymbol{\theta}}^b}^b(a) \, da$$

$$= 0.$$

For any $\boldsymbol{\theta}^w$, to show $\nabla_{\boldsymbol{\theta}^m} J(\pi_{\tilde{\boldsymbol{\theta}}^m}^m) = 0$, we can show $\nabla_{\boldsymbol{\theta}_k^b} J(\pi_{\tilde{\boldsymbol{\theta}}^m}^m) = 0$ and $\nabla_{\boldsymbol{\theta}^w} J(\pi_{\tilde{\boldsymbol{\theta}}^m}^m) = 0$.

We first derive the gradient of $\pi_{\boldsymbol{\theta}^m}^m$ with respect to $\boldsymbol{\theta}_k^b$ and $\boldsymbol{\theta}^w$:

$$\nabla_{\boldsymbol{\theta}_k^b} \pi_{\boldsymbol{\theta}^m}^m(a) = \nabla_{\boldsymbol{\theta}_k^b} \sum_{k=1}^N \pi_{\boldsymbol{\theta}^w}^w(k) \pi_{\boldsymbol{\theta}_k^b}^b(a) = \pi_{\boldsymbol{\theta}^w}^w(k) \nabla_{\boldsymbol{\theta}_k^b} \pi_{\boldsymbol{\theta}_k^b}^b(a),$$

$$\nabla_{\boldsymbol{\theta}^w} \pi_{\boldsymbol{\theta}^m}^m(a) = \nabla_{\boldsymbol{\theta}^w} \sum_{k=1}^N \pi_{\boldsymbol{\theta}^w}^w(k) \pi_{\boldsymbol{\theta}_k^b}^b(a) = \sum_{k=1}^N \nabla_{\boldsymbol{\theta}^w} \pi_{\boldsymbol{\theta}^w}^w(k) \pi_{\boldsymbol{\theta}_k^b}^b(a).$$

Then, we can derive the gradient $J(\pi_{\boldsymbol{\theta}^m}^m)$ with respect to $\boldsymbol{\theta}_k^b$:

$$\nabla_{\boldsymbol{\theta}_k^b} J(\pi_{\boldsymbol{\theta}^m}^m) = \nabla_{\boldsymbol{\theta}_k^b} \mathbb{E}_{a \sim \pi_{\boldsymbol{\theta}^m}^m(a)} \left[ r(a) - \alpha \log \pi_{\boldsymbol{\theta}^m}^m(a) \right]$$

$$= \nabla_{\boldsymbol{\theta}_k^b} \int_a \left( r(a) - \alpha \pi_{\boldsymbol{\theta}^m}^m(a) \log \pi_{\boldsymbol{\theta}^m}^m(a) \right) da$$

$$= \int_a \left( r(a) - \alpha \log \pi_{\boldsymbol{\theta}^m}^m(a) - \alpha \right) \nabla_{\boldsymbol{\theta}_k^b} \pi_{\boldsymbol{\theta}^m}^m(a) \, da$$

$$= \int_a \left( r(a) - \alpha \log \pi_{\boldsymbol{\theta}^m}^m(a) - \alpha \right) \pi_{\boldsymbol{\theta}^w}^w(k) \nabla_{\boldsymbol{\theta}_k^b} \pi_{\boldsymbol{\theta}_k^b}^b(a) \, da$$

$$= \pi_{\boldsymbol{\theta}^w}^w(k) \int_a \left( r(a) - \alpha \log \pi_{\boldsymbol{\theta}^m}^m(a) - \alpha \right) \nabla_{\boldsymbol{\theta}_k^b} \pi_{\boldsymbol{\theta}_k^b}^b(a) \, da.$$

Plugging $\tilde{\boldsymbol{\theta}}^m = \left[ \tilde{\boldsymbol{\theta}}^{b\top}, \cdots, \tilde{\boldsymbol{\theta}}^{b\top}, \boldsymbol{\theta}^{w\top} \right]^\top$ and $\pi_{\tilde{\boldsymbol{\theta}}^m}^m(a) = \pi_{\tilde{\boldsymbol{\theta}}^b}^b(a)$ in, we have

$$\nabla_{\boldsymbol{\theta}_k^b} J(\pi_{\tilde{\boldsymbol{\theta}}^m}^m) = \pi_{\boldsymbol{\theta}^w}^w(k) \int_a \left( r(a) - \alpha \log \pi_{\tilde{\boldsymbol{\theta}}^b}^b(a) - \alpha \right) \nabla_{\boldsymbol{\theta}_k^b} \pi_{\tilde{\boldsymbol{\theta}}^b}^b(a) \, da = 0.$$

Next, we derive the gradient $J(\pi_{\boldsymbol{\theta}^m}^m)$ with respect to $\boldsymbol{\theta}^w$:

$$\nabla_{\boldsymbol{\theta}^w} J(\pi_{\boldsymbol{\theta}^m}^m) = \nabla_{\boldsymbol{\theta}^w} \mathbb{E}_{a \sim \pi_{\boldsymbol{\theta}^m}^m(a)} \left[ r(a) - \alpha \log \pi_{\boldsymbol{\theta}^m}^m(a) \right]$$

$$= \nabla_{\boldsymbol{\theta}^w} \int_a \left( r(a) - \alpha \pi_{\boldsymbol{\theta}^m}^m(a) \log \pi_{\boldsymbol{\theta}^m}^m(a) \right) da$$

$$= \int_a \left( r(a) - \alpha \log \pi_{\boldsymbol{\theta}^m}^m(a) - \alpha \right) \nabla_{\boldsymbol{\theta}^w} \pi_{\boldsymbol{\theta}^m}^m(a) \, da$$

$$= \int_a \left( r(a) - \alpha \log \pi_{\boldsymbol{\theta}^m}^m(a) - \alpha \right) \sum_{k=1}^N \nabla_{\boldsymbol{\theta}^w} \pi_{\boldsymbol{\theta}^w}^w(k) \pi_{\boldsymbol{\theta}_k^b}^b(a) \, da.$$

Again, plugging $\tilde{\boldsymbol{\theta}}^m = \left[ \tilde{\boldsymbol{\theta}}^{b\top}, \cdots, \tilde{\boldsymbol{\theta}}^{b\top}, \boldsymbol{\theta}^{w\top} \right]^\top$ and $\pi_{\tilde{\boldsymbol{\theta}}^m}^m(a) = \pi_{\tilde{\boldsymbol{\theta}}^b}^b(a)$ in, we have

$$\nabla_{\boldsymbol{\theta}^w} J(\pi_{\tilde{\boldsymbol{\theta}}^m}^m) = \int_a \left( r(a) - \alpha \log \pi_{\tilde{\boldsymbol{\theta}}^b}^b(a) - \alpha \right) \sum_{k=1}^N \nabla_{\boldsymbol{\theta}^w} \pi_{\boldsymbol{\theta}^w}^w(k) \pi_{\tilde{\boldsymbol{\theta}}^b}^b(a) \, da$$

$$= \int_a \left( r(a) - \alpha \log \pi_{\tilde{\boldsymbol{\theta}}^b}^b(a) - \alpha \right) \pi_{\tilde{\boldsymbol{\theta}}^b}^b(a) \, da \sum_{k=1}^N \nabla_{\boldsymbol{\theta}^w} \pi_{\boldsymbol{\theta}^w}^w(k)$$

$$= \int_a \left( r(a) - \alpha \log \pi_{\tilde{\boldsymbol{\theta}}^b}^b(a) - \alpha \right) \pi_{\tilde{\boldsymbol{\theta}}^b}^b(a) \, da \nabla_{\boldsymbol{\theta}^w} \sum_{k=1}^N \pi_{\boldsymbol{\theta}^w}^w(k)$$

$$= \int_a \left( r(a) - \alpha \log \pi_{\tilde{\boldsymbol{\theta}}^b}^b(a) - \alpha \right) \pi_{\tilde{\boldsymbol{\theta}}^b}^b(a) \, da \nabla_{\boldsymbol{\theta}^w} 1$$

$$= 0.$$

Thus, $\nabla_{\tilde{\boldsymbol{\theta}}} J(\pi_{\tilde{\boldsymbol{\theta}}^m}^m) = \left[ \nabla_{\boldsymbol{\theta}_1^b} J(\pi_{\tilde{\boldsymbol{\theta}}^m}^m)^\top, \cdots, \nabla_{\boldsymbol{\theta}_N^b} J(\pi_{\tilde{\boldsymbol{\theta}}^m}^m)^\top, \nabla_{\boldsymbol{\theta}^w} J(\pi_{\tilde{\boldsymbol{\theta}}^m}^m)^\top \right]^\top = \mathbf{0}$. $\qquad \square$

### B.2 Proofs for Results in Section 4

Define the (discounted) occupancy measure under $\pi_{\boldsymbol{\theta}}^m$ as $d_{\pi_{\boldsymbol{\theta}}^m}(s) \doteq \sum_{t=0}^\infty \mathbb{E}_{\pi_{\boldsymbol{\theta}}^m} [\gamma^t \mathbb{I}(S_t = s)]$. We first prove the half-reparameterization policy gradient theorem, which is a special case of Theorem 4.3 with $\alpha = 0$.

**Assumption 4.1.** $\mathcal{S}$ and $\mathcal{A}$ are compact.

**Assumption 4.2.** $p(s'|s,a)$, $d_0(s)$, $r(s,a)$ $f_{\boldsymbol{\theta}}(\epsilon; s, k)$, $f_{\boldsymbol{\theta}}^{-1}(a; s, k)$, $\pi_{\boldsymbol{\theta}}^w(k|s)$, $\pi_{\boldsymbol{\theta}}^b(a|s,k)$, $p(\epsilon)$, and their derivatives are continuous in variables $s$, $a$, $s'$, $\boldsymbol{\theta}$, and $\epsilon$.

**Theorem B.1** (Half-Reparameterization Policy Gradient Theorem). *Under Assumptions 4.1 and 4.2, we have*

$$\nabla_{\boldsymbol{\theta}} J_0(\pi_{\boldsymbol{\theta}}^m) = \mathbb{E}_{s \sim d_{\pi_{\boldsymbol{\theta}}^m}, k \sim \pi_{\boldsymbol{\theta}}^w(\cdot|s), \epsilon \sim p} \Big[ Q_{\pi_{\boldsymbol{\theta}}^m}(s, f_{\boldsymbol{\theta}}(\epsilon; s, k)) \nabla_{\boldsymbol{\theta}} \log \pi_{\boldsymbol{\theta}}^w(k|s)$$
$$+ \nabla_{\boldsymbol{\theta}} f_{\boldsymbol{\theta}}(\epsilon; s, k) \nabla_a Q_{\pi_{\boldsymbol{\theta}}^m}(s, a)|_{a=f_{\boldsymbol{\theta}}(\epsilon; s, k)} \Big].$$

*Proof.* We start with the policy gradient theorem (Sutton et al., 1999), which shows

$$\nabla_{\boldsymbol{\theta}} J_0(\pi_{\boldsymbol{\theta}}^m) = \int_{s,a} d_{\pi_{\boldsymbol{\theta}}^m}(s) \pi_{\boldsymbol{\theta}}^m(a|s) Q_{\pi_{\boldsymbol{\theta}}^m}(s, a) \nabla_{\boldsymbol{\theta}} \log \pi_{\boldsymbol{\theta}}^m(a|s) \, da \, ds.$$

Then

$$\nabla_{\boldsymbol{\theta}} J_0(\pi_{\boldsymbol{\theta}}^m) = \int_{s,a} d_{\pi_{\boldsymbol{\theta}}^m}(s) \pi_{\boldsymbol{\theta}}^m(a|s) Q_{\pi_{\boldsymbol{\theta}}^m}(s, a) \nabla_{\boldsymbol{\theta}} \log \pi_{\boldsymbol{\theta}}^m(a|s) \, da \, ds$$

$$= \int_s d_{\pi_{\boldsymbol{\theta}}^m}(s) \left( \int_a Q_{\pi_{\boldsymbol{\theta}}^m}(s, a) \nabla_{\boldsymbol{\theta}} \pi_{\boldsymbol{\theta}}^m(a|s) \, da \right) ds \tag{14}$$

$$= \int_s d_{\pi_{\boldsymbol{\theta}}^m}(s) \left( \int_a Q_{\pi_{\boldsymbol{\theta}}^m}(s, a) \nabla_{\boldsymbol{\theta}} \left( \sum_{k=1}^N \pi_{\boldsymbol{\theta}}^w(k|s) \pi_{\boldsymbol{\theta}}^b(a|s, k) \right) da \right) ds$$

$$= \int_s d_{\pi_{\boldsymbol{\theta}}^m}(s) \sum_{k=1}^N \left( \int_a Q_{\pi_{\boldsymbol{\theta}}^m}(s, a) \nabla_{\boldsymbol{\theta}} \left( \pi_{\boldsymbol{\theta}}^w(k|s) \pi_{\boldsymbol{\theta}}^b(a|s, k) \right) da \right) ds$$

$$= \int_s d_{\pi_{\boldsymbol{\theta}}^m}(s) \sum_{k=1}^N \left( \int_a Q_{\pi_{\boldsymbol{\theta}}^m}(s, a) \nabla_{\boldsymbol{\theta}} \pi_{\boldsymbol{\theta}}^w(k|s) \pi_{\boldsymbol{\theta}}^b(a|s, k) \, da \right.$$

$$\left. + \int_a Q_{\pi_{\boldsymbol{\theta}}^m}(s, a) \pi_{\boldsymbol{\theta}}^w(k|s) \nabla_{\boldsymbol{\theta}} \pi_{\boldsymbol{\theta}}^b(a|s, k) \, da \right) ds$$

$$= \int_s d_{\pi_{\boldsymbol{\theta}}^m}(s) \sum_{k=1}^N \left( \int_a Q_{\pi_{\boldsymbol{\theta}}^m}(s, a) \nabla_{\boldsymbol{\theta}} \log \pi_{\boldsymbol{\theta}}^w(k|s) \pi_{\boldsymbol{\theta}}^w(k|s) \pi_{\boldsymbol{\theta}}^b(a|s, k) \, da \right.$$

$$\left. + \pi_{\boldsymbol{\theta}}^w(k|s) \left( \int_a \nabla_{\boldsymbol{\theta}} \left( Q_{\pi_{\boldsymbol{\theta}}^m}(s, a) \pi_{\boldsymbol{\theta}}^b(a|s, k) \right) da - \int_a \pi_{\boldsymbol{\theta}}^b(a|s, k) \nabla_{\boldsymbol{\theta}} Q_{\pi_{\boldsymbol{\theta}}^m}(s, a) \, da \right) \right) ds$$

$$= \int_s d_{\pi_{\boldsymbol{\theta}}^m}(s) \sum_{k=1}^N \pi_{\boldsymbol{\theta}}^w(k|s) \left( \int_a Q_{\pi_{\boldsymbol{\theta}}^m}(s, a) \nabla_{\boldsymbol{\theta}} \log \pi_{\boldsymbol{\theta}}^w(k|s) \pi_{\boldsymbol{\theta}}^b(a|s, k) \, da \right.$$

$$\left. + \nabla_{\boldsymbol{\theta}} \int_a Q_{\pi_{\boldsymbol{\theta}}^m}(s, a) \pi_{\boldsymbol{\theta}}^b(a|s, k) \, da - \int_a \pi_{\boldsymbol{\theta}}^b(a|s, k) \nabla_{\boldsymbol{\theta}} Q_{\pi_{\boldsymbol{\theta}}^m}(s, a) \, da \right) ds$$

$$\overset{a=f_{\boldsymbol{\theta}}(\epsilon; s, k)}{=} \int_s d_{\pi_{\boldsymbol{\theta}}^m}(s) \sum_{k=1}^N \pi_{\boldsymbol{\theta}}^w(k|s) \left( \int_\epsilon p(\epsilon) Q_{\pi_{\boldsymbol{\theta}}^m}(s, f_{\boldsymbol{\theta}}(\epsilon; s, k)) \nabla_{\boldsymbol{\theta}} \log \pi_{\boldsymbol{\theta}}^w(k|s) \, d\epsilon \right.$$

$$\left. + \nabla_{\boldsymbol{\theta}} \int_\epsilon p(\epsilon) Q_{\pi_{\boldsymbol{\theta}}^m}(s, f_{\boldsymbol{\theta}}(\epsilon; s, k)) \, d\epsilon - \int_\epsilon p(\epsilon) \nabla_{\boldsymbol{\theta}} Q_{\pi_{\boldsymbol{\theta}}^m}(s, a)|_{a=f_{\boldsymbol{\theta}}(\epsilon; s, k)} \, d\epsilon \right) ds$$

$$= \int_s d_{\pi_{\boldsymbol{\theta}}^m}(s) \sum_{k=1}^N \pi_{\boldsymbol{\theta}}^w(k|s) \left( \int_\epsilon p(\epsilon) Q_{\pi_{\boldsymbol{\theta}}^m}(s, f_{\boldsymbol{\theta}}(\epsilon; s, k)) \nabla_{\boldsymbol{\theta}} \log \pi_{\boldsymbol{\theta}}^w(k|s) \, d\epsilon \right.$$

$$\left. + \int_\epsilon p(\epsilon) \nabla_{\boldsymbol{\theta}} f_{\boldsymbol{\theta}}(\epsilon; s, k) \nabla_a Q_{\pi_{\boldsymbol{\theta}}^m}(s, a)|_{a=f_{\boldsymbol{\theta}}(\epsilon; s, k)} \, d\epsilon \right) ds$$

$$= \mathbb{E}_{s \sim d_{\pi_{\boldsymbol{\theta}}^m}, k \sim \pi_{\boldsymbol{\theta}}^w(\cdot|s), \epsilon \sim p} \Big[ Q_{\pi_{\boldsymbol{\theta}}^m}(s, f_{\boldsymbol{\theta}}(\epsilon; s, k)) \nabla_{\boldsymbol{\theta}} \log \pi_{\boldsymbol{\theta}}^w(k|s)$$

$$+ \nabla_{\boldsymbol{\theta}} f_{\boldsymbol{\theta}}(\epsilon; s, k) \nabla_a Q_{\pi_{\boldsymbol{\theta}}^m}(s, a)|_{a=f_{\boldsymbol{\theta}}(\epsilon; s, k)} \Big],$$

where the second last equality is due to

$$\nabla_{\boldsymbol{\theta}} Q_{\pi_{\boldsymbol{\theta}}^m}(s, f_{\boldsymbol{\theta}}(\epsilon; s, k)) - \nabla_{\boldsymbol{\theta}} Q_{\pi_{\boldsymbol{\theta}}^m}(s, a)|_{a=f_{\boldsymbol{\theta}}(\epsilon; s, k)}$$

$$= \nabla_{\boldsymbol{\theta}} f_{\boldsymbol{\theta}}(\epsilon; s, k) \nabla_a Q_{\pi_{\boldsymbol{\theta}}^m}(s, a)|_{a=f_{\boldsymbol{\theta}}(\epsilon; s, k)} + \nabla_{\boldsymbol{\theta}} Q_{\pi_{\boldsymbol{\theta}}^m}(s, a)|_{a=f_{\boldsymbol{\theta}}(\epsilon; s, k)} - \nabla_{\boldsymbol{\theta}} Q_{\pi_{\boldsymbol{\theta}}^m}(s, a)|_{a=f_{\boldsymbol{\theta}}(\epsilon; s, k)}$$

$$= \nabla_{\boldsymbol{\theta}} f_{\boldsymbol{\theta}}(\epsilon; s, k) \nabla_a Q_{\pi_{\boldsymbol{\theta}}^m}(s, a)|_{a=f_{\boldsymbol{\theta}}(\epsilon; s, k)}.$$

□

*Remark* B.2. The key contribution of this proof is the decoupling of the gradient of the weighting policy $\pi_{\boldsymbol{\theta}}^w$ and the gradient of the component policies $\pi_{\boldsymbol{\theta}}^b$. The former, $\nabla_{\boldsymbol{\theta}} \pi_{\boldsymbol{\theta}}^w$, is converted back to the likelihood-ratio gradient, while the latter, $\nabla_{\boldsymbol{\theta}} \pi_{\boldsymbol{\theta}}^b$, is handled in the same way as in the proof the reparameterization policy gradient theorem (Lan et al., 2022).

Combining the insight from Remark B.2 with the proof of the entropy-regularized reparameterization policy gradient theorem in Lan et al. (2022), we can obtain Theorem B.3, which is a restatement of Theorem 4.3.

**Theorem B.3** (Entropy-Regularized Half-Reparameterization Policy Gradient Theorem). *Under Assumptions 4.1 and 4.2, we have*

$$\nabla_{\boldsymbol{\theta}} J(\pi_{\boldsymbol{\theta}}^m) = \mathbb{E}_{s \sim d_{\pi_{\boldsymbol{\theta}}^m}, k \sim \pi_{\boldsymbol{\theta}}^w(\cdot|s), \epsilon \sim p} \Big[ \nabla_{\boldsymbol{\theta}} \log \pi_{\boldsymbol{\theta}}^w(k|s) \big( Q_{\pi_{\boldsymbol{\theta}}^m}(s, f_{\boldsymbol{\theta}}(\epsilon; s, k)) - \alpha \log \pi_{\boldsymbol{\theta}}^m(f_{\boldsymbol{\theta}}(\epsilon; s, k)|s) \big)$$

$$+ \nabla_{\boldsymbol{\theta}} f_{\boldsymbol{\theta}}(\epsilon; s, k) \nabla_a \big( Q_{\pi_{\boldsymbol{\theta}}^m}(s, a) - \alpha \log \pi_{\boldsymbol{\theta}}^m(a|s) \big)|_{a=f_{\boldsymbol{\theta}}(\epsilon; s, k)} \Big].$$

*Proof.* From (3) of Ahmed et al. (2019), we have the entropy-regularized policy gradient for the regularized objective:

$$\nabla_{\boldsymbol{\theta}} J(\pi_{\boldsymbol{\theta}}^m) = \int_{s, a} d_{\pi_{\boldsymbol{\theta}}^m}(s) \pi_{\boldsymbol{\theta}}^m(a|s) \big( Q_{\pi_{\boldsymbol{\theta}}^m}(s, a) \nabla_{\boldsymbol{\theta}} \log \pi_{\boldsymbol{\theta}}^m(a|s) + \alpha \nabla_{\boldsymbol{\theta}} \mathcal{H}(\pi_{\boldsymbol{\theta}}^m(\cdot|s)) \big) \, da \, ds.$$

The first term, $Q_{\pi_{\boldsymbol{\theta}}^m}(s, a) \nabla_{\boldsymbol{\theta}} \log \pi_{\boldsymbol{\theta}}^m(a|s)$, can be directly handled by Theorem B.1. Here, we analyze the second term, $\alpha \nabla_{\boldsymbol{\theta}} \mathcal{H}(\pi_{\boldsymbol{\theta}}^m(\cdot|s))$. Notice that

$$\nabla_{\boldsymbol{\theta}} \mathcal{H}(\pi_{\boldsymbol{\theta}}^m(\cdot|s)) = -\nabla_{\boldsymbol{\theta}} \int_a \pi_{\boldsymbol{\theta}}^m(a|s) \log \pi_{\boldsymbol{\theta}}^m(a|s) \, da$$

$$= - \int_a \big( \nabla_{\boldsymbol{\theta}} \pi_{\boldsymbol{\theta}}^m(a|s) \log \pi_{\boldsymbol{\theta}}^m(a|s) + \pi_{\boldsymbol{\theta}}^m(a|s) \nabla_{\boldsymbol{\theta}} \log \pi_{\boldsymbol{\theta}}^m(a|s) \big) \, da$$

$$= - \int_a \big( \nabla_{\boldsymbol{\theta}} \pi_{\boldsymbol{\theta}}^m(a|s) \log \pi_{\boldsymbol{\theta}}^m(a|s) + \nabla_{\boldsymbol{\theta}} \pi_{\boldsymbol{\theta}}^m(a|s) \big) \, da$$

$$\overset{\int_a \nabla_{\boldsymbol{\theta}} \pi_{\boldsymbol{\theta}}^m(a|s) \, da = 0}{=} - \int_a \nabla_{\boldsymbol{\theta}} \pi_{\boldsymbol{\theta}}^m(a|s) \log \pi_{\boldsymbol{\theta}}^m(a|s) \, da,$$

then we have

$$\int_{s, a} d_{\pi_{\boldsymbol{\theta}}^m}(s) \pi_{\boldsymbol{\theta}}^m(a|s) \alpha \nabla_{\boldsymbol{\theta}} \mathcal{H}(\pi_{\boldsymbol{\theta}}^m(\cdot|s)) \, da \, ds$$

$$= \alpha \int_s d_{\pi_{\boldsymbol{\theta}}^m}(s) \nabla_{\boldsymbol{\theta}} \mathcal{H}(\pi_{\boldsymbol{\theta}}^m(\cdot|s)) \, ds$$

$$= - \alpha \int_s d_{\pi_{\boldsymbol{\theta}}^m}(s) \int_a \nabla_{\boldsymbol{\theta}} \pi_{\boldsymbol{\theta}}^m(a|s) \log \pi_{\boldsymbol{\theta}}^m(a|s) \, da \, ds. \tag{15}$$

Since (15) resembles (14), by following the same steps in the proof of Theorem B.1, we can obtain

$$\int_{s, a} d_{\pi_{\boldsymbol{\theta}}^m}(s) \pi_{\boldsymbol{\theta}}^m(a|s) \alpha \nabla_{\boldsymbol{\theta}} \mathcal{H}(\pi_{\boldsymbol{\theta}}^m(\cdot|s)) \, da \, ds$$

$$= \mathbb{E}_{s \sim d_{\pi_{\boldsymbol{\theta}}^m}, k \sim \pi_{\boldsymbol{\theta}}^w(\cdot|s), \epsilon \sim p} \Big[ - \alpha \log \pi_{\boldsymbol{\theta}}^m(f_{\boldsymbol{\theta}}(\epsilon; s, k)|s) \nabla_{\boldsymbol{\theta}} \log \pi_{\boldsymbol{\theta}}^w(k|s)$$

$$- \alpha \nabla_{\boldsymbol{\theta}} f_{\boldsymbol{\theta}}(\epsilon; s, k) \nabla_a \log \pi_{\boldsymbol{\theta}}^m(a|s)|_{a=f_{\boldsymbol{\theta}}(\epsilon; s, k)} \Big],$$

Combining the above gradient term with the gradient term from Theorem B.1 concludes the proof.

□

By using the same technique, we can obtain the half-reparameterization gradient of SAC's objective in (3).

**Assumption 4.4.** $Q_{\mathbf{w}}(s, a)$ and its derivatives are continuous in variables $s$ and $a$.

**Proposition B.4.** *Under Assumptions 4.1, 4.2, and 4.4, we have*

$$
\nabla_{\boldsymbol{\theta}} \hat{J}(\pi_{\boldsymbol{\theta}}^w) = \mathbb{E}_{s \sim \mathcal{B}, k \sim \pi_{\boldsymbol{\theta}}^w(\cdot|s), \epsilon \sim p} \Big[ \nabla_{\boldsymbol{\theta}} \log \pi_{\boldsymbol{\theta}}^w(k|s) \big( Q_{\mathbf{w}}(s, f_{\boldsymbol{\theta}}(\epsilon; s, k)) - \alpha \log \pi_{\boldsymbol{\theta}}^m(f_{\boldsymbol{\theta}}(\epsilon; s, k)|s)) \\
+ \nabla_{\boldsymbol{\theta}} \big( Q_{\mathbf{w}}(s, f_{\boldsymbol{\theta}}(\epsilon; s, k)) - \alpha \log \pi_{\boldsymbol{\theta}}^m(f_{\boldsymbol{\theta}}(\epsilon; s, k)|s)) \Big].
$$

*Proof.* We first rewrite (3) with reparameterized component policies:

$$
\hat{J}(\pi_{\boldsymbol{\theta}}^w) = \mathbb{E}_{S_t \sim \mathcal{B}, A_t \sim \pi_{\boldsymbol{\theta}}^m} \left[ Q_{\mathbf{w}}(S_t, A_t) - \alpha \log \pi_{\boldsymbol{\theta}}^m(A_t|S_t) \right]
$$

$$
= \int_s d_{\mathcal{B}}(s) \int_a \pi_{\boldsymbol{\theta}}^m(a|s) \big( Q_{\mathbf{w}}(s, a) - \alpha \log \pi_{\boldsymbol{\theta}}^m(a|s) \big) \, da \, ds
$$

$$
= \int_s d_{\mathcal{B}}(s) \int_a \sum_{k=1}^{N} \pi_{\boldsymbol{\theta}}^w(k|s) \pi_{\boldsymbol{\theta}}^b(a|s, k) \big( Q_{\mathbf{w}}(s, a) - \alpha \log \pi_{\boldsymbol{\theta}}^m(a|s) \big) \, da \, ds
$$

$$
\stackrel{a = f_{\boldsymbol{\theta}}(\epsilon; s, k)}{=} \int_s d_{\mathcal{B}}(s) \int_\epsilon \sum_{k=1}^{N} \pi_{\boldsymbol{\theta}}^w(k|s) p(\epsilon) \big( Q_{\mathbf{w}}(s, f_{\boldsymbol{\theta}}(\epsilon; s, k)) - \alpha \log \pi_{\boldsymbol{\theta}}^m(f_{\boldsymbol{\theta}}(\epsilon; s, k)|s) \big) \, d\epsilon \, ds
$$

$$
= \int_s d_{\mathcal{B}}(s) \int_\epsilon p(\epsilon) \sum_{k=1}^{N} \pi_{\boldsymbol{\theta}}^w(k|s) \big( Q_{\mathbf{w}}(s, f_{\boldsymbol{\theta}}(\epsilon; s, k)) - \alpha \log \pi_{\boldsymbol{\theta}}^m(f_{\boldsymbol{\theta}}(\epsilon; s, k)|s) \big) \, d\epsilon \, ds.
$$

We can then derive its gradient:

$$
\nabla_{\boldsymbol{\theta}} \hat{J}(\pi_{\boldsymbol{\theta}}^m)
$$

$$
= \nabla_{\boldsymbol{\theta}} \int_s d_{\mathcal{B}}(s) \int_\epsilon p(\epsilon) \sum_{k=1}^{N} \pi_{\boldsymbol{\theta}}^w(k|s) \big( Q_{\mathbf{w}}(s, f_{\boldsymbol{\theta}}(\epsilon; s, k)) - \alpha \log \pi_{\boldsymbol{\theta}}^m(f_{\boldsymbol{\theta}}(\epsilon; s, k)|s) \big) \, d\epsilon \, ds
$$

$$
= \int_s d_{\mathcal{B}}(s) \int_\epsilon p(\epsilon) \sum_{k=1}^{N} \nabla_{\boldsymbol{\theta}} \Big( \pi_{\boldsymbol{\theta}}^w(k|s) \big( Q_{\mathbf{w}}(s, f_{\boldsymbol{\theta}}(\epsilon; s, k)) - \alpha \log \pi_{\boldsymbol{\theta}}^m(f_{\boldsymbol{\theta}}(\epsilon; s, k)|s) \big) \Big) \, d\epsilon \, ds
$$

$$
= \int_s d_{\mathcal{B}}(s) \int_\epsilon p(\epsilon) \sum_{k=1}^{N} \Big( \nabla_{\boldsymbol{\theta}} \pi_{\boldsymbol{\theta}}^w(k|s) \big( Q_{\mathbf{w}}(s, f_{\boldsymbol{\theta}}(\epsilon; s, k)) - \alpha \log \pi_{\boldsymbol{\theta}}^m(f_{\boldsymbol{\theta}}(\epsilon; s, k)|s) \big)
$$

$$
+ \pi_{\boldsymbol{\theta}}^w(k|s) \nabla_{\boldsymbol{\theta}} \big( Q_{\mathbf{w}}(s, f_{\boldsymbol{\theta}}(\epsilon; s, k)) - \alpha \log \pi_{\boldsymbol{\theta}}^m(f_{\boldsymbol{\theta}}(\epsilon; s, k)|s) \big) \Big) \, d\epsilon \, ds
$$

$$
= \int_s d_{\mathcal{B}}(s) \int_\epsilon p(\epsilon) \sum_{k=1}^{N} \Big( \pi_{\boldsymbol{\theta}}^w(k|s) \nabla_{\boldsymbol{\theta}} \log \pi_{\boldsymbol{\theta}}^w(k|s) \big( Q_{\mathbf{w}}(s, f_{\boldsymbol{\theta}}(\epsilon; s, k)) - \alpha \log \pi_{\boldsymbol{\theta}}^m(f_{\boldsymbol{\theta}}(\epsilon; s, k)|s) \big)
$$

$$
+ \pi_{\boldsymbol{\theta}}^w(k|s) \nabla_{\boldsymbol{\theta}} \big( Q_{\mathbf{w}}(s, f_{\boldsymbol{\theta}}(\epsilon; s, k)) - \alpha \log \pi_{\boldsymbol{\theta}}^m(f_{\boldsymbol{\theta}}(\epsilon; s, k)|s) \big) \Big) \, d\epsilon \, ds
$$

$$
= \int_s d_{\mathcal{B}}(s) \int_\epsilon p(\epsilon) \sum_{k=1}^{N} \pi_{\boldsymbol{\theta}}^w(k|s) \Big( \nabla_{\boldsymbol{\theta}} \log \pi_{\boldsymbol{\theta}}^w(k|s) \big( Q_{\mathbf{w}}(s, f_{\boldsymbol{\theta}}(\epsilon; s, k)) - \alpha \log \pi_{\boldsymbol{\theta}}^m(f_{\boldsymbol{\theta}}(\epsilon; s, k)|s) \big)
$$

$$
+ \nabla_{\boldsymbol{\theta}} \big( Q_{\mathbf{w}}(s, f_{\boldsymbol{\theta}}(\epsilon; s, k)) - \alpha \log \pi_{\boldsymbol{\theta}}^m(f_{\boldsymbol{\theta}}(\epsilon; s, k)|s) \big) \Big) \, d\epsilon \, ds
$$

$$
= \mathbb{E}_{s \sim \mathcal{B}, k \sim \pi_{\boldsymbol{\theta}}^w(\cdot|s), \epsilon \sim p} \Big[ \nabla_{\boldsymbol{\theta}} \log \pi_{\boldsymbol{\theta}}^w(k|s) \big( Q_{\mathbf{w}}(s, f_{\boldsymbol{\theta}}(\epsilon; s, k)) - \alpha \log \pi_{\boldsymbol{\theta}}^m(f_{\boldsymbol{\theta}}(\epsilon; s, k)|s) \big)
$$

$$
+ \nabla_{\boldsymbol{\theta}} \big( Q_{\mathbf{w}}(s, f_{\boldsymbol{\theta}}(\epsilon; s, k)) - \alpha \log \pi_{\boldsymbol{\theta}}^m(f_{\boldsymbol{\theta}}(\epsilon; s, k)|s) \big) \Big].
$$

$\square$

# C  NUMERICAL STUDY DETAILS

In this section, we provide more details on the numerical study presented in Section 3.3. The reward function of the bimodal bandit is the normalized summation of two Gaussians' density functions whose standard deviations are both $0.5$ and whose means are $-1$ and $1$, respectively. We use the default optimization algorithm for variable with bounds in SciPy (Virtanen et al., 2020) to optimize the entropy regularized objective $J(\pi_{\boldsymbol{\theta}}) = \mathbb{E}_{a \sim \pi_{\boldsymbol{\theta}}}[r(a) - \alpha \log \pi_{\boldsymbol{\theta}}(a)]$, where $r(a)$ is the value of the action depicted in Figure 1. We then sort the obtained stationary points based on their regularized values $J(\pi_{\boldsymbol{\theta}})$ and use the ones that have the highest values as the parameters of optimal policies for Figure 1.

For each policy class, we run the default optimization algorithm for 100 trials, each with a set of randomly sampled initial policy parameters. Specifically, the initial means, log standard deviations, and mixing weights are randomly sampled from $[-2, 2]$, $[-3, 0]$, and $[0, 1]$, respectively. To avoid numerical issues in numerical integral when the standard deviation gets too large, we impose an upper bound of 3 for the log standard deviation. In addition, the mixing weights are defined and bounded within $[0, 1]$. We initially run this optimization procedure for seven different entropy scales: $\alpha \in \{0.05, 0.1, 0.2, 0.3, 0.4, 0.5, 0.6\}$. With difference shown when $\alpha$ is between $0.2$ and $0.5$ (see Figure 1 (Middle)), we additionally run another eight entropy scales: $\alpha \in \{0.22, 0.24, 0.26, 0.28, 0.325, 0.35, 0.375, 0.45\}$ to obtain more insights when $\alpha$ within this range. Note that, we didn't obtain any convergent results for the Gaussian policy when $\alpha >= 0.325$ and for the Gaussian mixture (GM) policy when $\alpha > 0.5$.

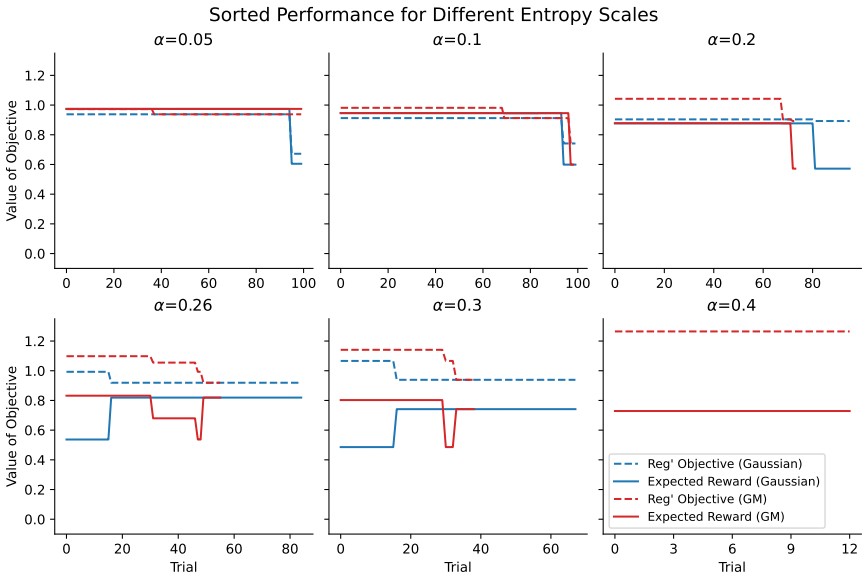

Figure 7: Expected reward and objective value of the regularized objective's stationary points found in 100 trials for the Gaussian and the Gaussian mixture policies. Note that the stationary points are sorted by their regularized objective value for clarity (dashed line). When a line ends before Trial 100, it means that the rest of the trials either diverged or encountered a numerical issue. The difference between the dash line and the solid line represents the differential entropy scaled by $\alpha$.

Figure 7 shows the expected reward and objective value of all the stationary points found for representative $\alpha$. Apart from the observation discussed in Section 3.3, we can see that both the Gaussian and the GM mixture often have different types of stationary points. The optimal Gaussian policy concentrates on one of the reward modes with a high expected reward with small $\alpha$, but it then shifts to the middle of two modes in $\alpha = 0.26$ and $\alpha = 0.3$ with a lower expected reward but a much higher entropy. The optimal GM policy, on the other hand, always has two modes covering two reward modes: it obtains a high reward while maintaining a higher entropy.

# D EXPERIMENT DETAILS

## D.1 EXPERIMENTAL DETAILS

In this section, we supply the omitted experimental details in Section 5.

**Common configurations.** For SAC with a likelihood-ratio gradient for the actor, we sample $N = 30$ actions for the given state and use the average of the corresponding action values as the baseline (see Algorithm 1). We also use such a baseline for the likelihood-ratio part of the half-reparameterization gradient estimator. We use the Adam optimizer (Kingma & Ba, 2014) with $\beta_1 = 0.9$ and $\beta_2 = 0.999$ for all experiments.

**MuJoCo and DeepMind Control Suite environments.** We use a two-layer feedforward network with a hidden dimension of $256$, a replay buffer size of $1,000,000$, and a batch size of $100$. We use the automatic entropy tuning and the same initial step size $3 \times 10^{-4}$ for the actor, critic, and entropy scale. We use a double Q network and a target network for the critic, which is an exponential moving average of the critic with a smoothing factor of $0.005$. In the initial $10,000$ steps, the actions are uniformly sampled.

**Classic control environments.** We use a two-layer feedforward network with a hidden dimension of $64$, a replay buffer size of $100,000$, and a batch size of $32$. We use a double Q network and a target network for the critic, which is an exponential moving average of the critic with a smoothing factor of $0.01$.

**Multimodal bandits.** We use a two-layer feedforward network with a hidden dimension of $16$, a replay buffer size of $5000$, and a batch size of $32$. Note that we replace the critic with the true value function as these bandits with a deterministic reward function are for illustration purposes.

**Network architecture of mixture policies.** Compared to the base policy's actor, the mixture policy's actor has additional heads for the additional parameters in the mixture distribution. For example, the last layer of a squashed Gaussian policy's actor has two outputs (one for the mean and one for the standard deviation), while the last layer of a squashed Gaussian policy's actor with five components has 15 outputs (five for the means, five for the standard deviations, and five for the mixing weights).

**Reference training time.** We provide training time samples for Pendulum and HalfCheetah in Table 1 for reference. The training time samples is obtained via an example run when the server is idle and no other active program is running. The CPU of the server is AMD Ryzen 9 5900X 12-Core Processor, and the GPU of the server is NVIDIA Geforce RTX 3080 Ti.

Table 1: Training time sample.

|  | Rep-Squashed | HalfRep-SGM | GumbelRep-SGM |
|---|---|---|---|
| Pendulum-CPU-10K | 24s | 42s | 31s |
| Pendulum-CPU-100K (projected) | 4m | 7m | 5m10s |
| HalfCheetah-GPU-100K | 627s | 843s | 801s |
| HalfCheetah-GPU-1000K (projected) | 1h40m30s | 2h20m30s | 2h13m30s |

## D.2 CLASSIC CONTROL ENVIRONMENT DETAILS

We use the v1 version of Pendulum from OpenAI Gym for ShapedPendulum. For Pendulum, we set the reward to 1 if the angle of the pendulum from the upright position is smaller than 0.25 and 0 otherwise. For ShapedAcrobot, we set the reward to be $-\cos(\theta_1) - \cos(\theta_2 + \theta_1) - 1.0$, where $\theta_1$ and $\theta_2$ are the first two dimensions of the state. For ShapedMountainCar, we set the reward to be $x - 0.6$, where $x$ is first dimension of the state. For Acrobot and MountainCar, we adapt the discrete version in Gym to the continuous action case as it is done in Neumann et al. (2022). All

environments use a discount factor of 0.99. The episode cut-offs for the Pendulum, Acrobot, and MountainCar are 200, 1000, and 1000, respectively.

### D.3 PLOTS FOR REMAINING MUJOCO AND DEEPMIND CONTROL SUITE ENVIRONMENTS

In Figure 8, we present the learning curves in the remaining three MuJoCo and four DeepMind Control Suite environments. The conclusions are similar to those in the main text.

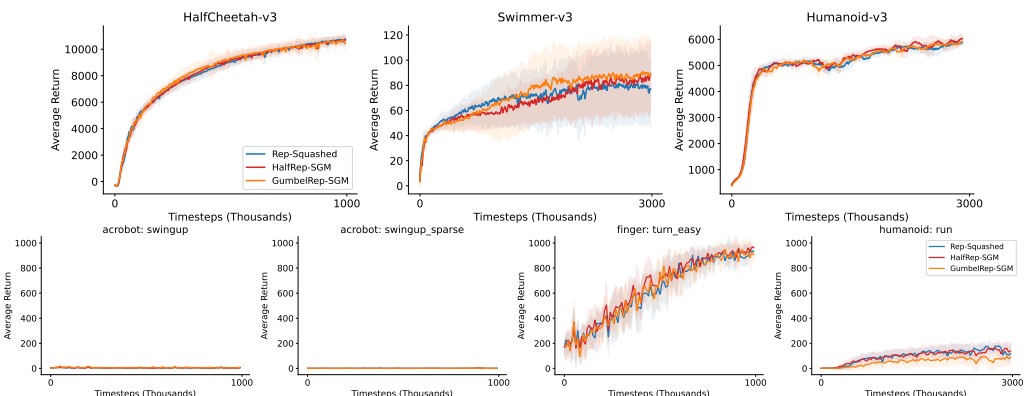

Figure 8: Learning curves in the remaining three MuJoCo and four DeepMind Control Suite environments. The shaded area shows the $95\%$ bootstrap confidence intervals across 8 runs.

### D.4 DETAILS FOR DATA VISUALIZATION

**Visualization of the action-value estimates and policy density for the base policy.** We presented the visualization of the action-value estimates and policy density for the mixture policy in Figure 5 to highlight the difference between ShapedMountainCar and MountainCar, one with shaped rewards and the other with unshaped rewards. In Figure 9, we show the same plot for the base policy, Rep-Squashed. The difference between the two types of environments is not so much different from that in Figure 5, but we can see the density is always unimodal during training, indicating less efficient exploration.

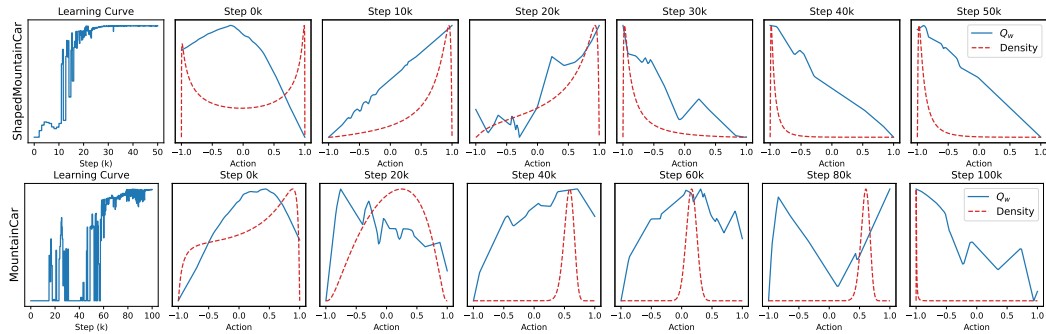

Figure 9: Learning curve, action-value estimates, and policy density at a starting state from a sample run of Rep-Squashed in two MountainCar variants. The y-axes differ across plots and are not shown with ticks to highlight the shape of the curves rather than their exact values. The observations are similar to those in Figure 5 with the exception that the density is always unimodal except for the starting step. It indicates that Rep-Squashed explore less efficiently, potentially explaining its worse performance in MountainCar.

## D.5 ADDITIONAL PLOTS FOR MULTIMODAL BANDITS EXPERIMENTS

Figure 10 shows the learning curves of Like-Squashed and Like-SGM with their best hyperparameter setting in all 100 multimodal bandits. We can see that Like-SGM outperforms Like-Squashed in a handful of bandits while tying with the latter in the rest.

### Results on Each Reward Function

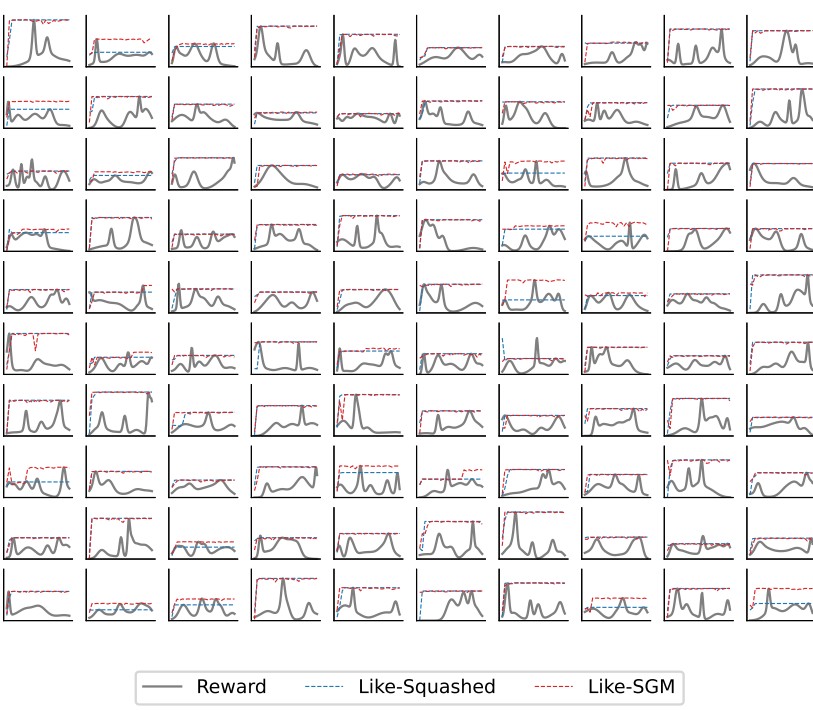

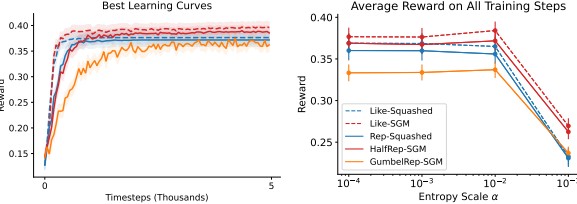

Figure 10: Learning curves of Like-Squashed and Like-SGM in each bandit. Each bandit is run for one seed.

Figure 11: Left: Best learning curves averaged over 100 bandits. Right: Average reward over all training steps. The shaded area and error bars show the 95% bootstrap confidence intervals across different bandits.

Figure 11 (Left) shows the learning curves of the hyperparameter setting that achieves the best final performance. We can see that the speed of convergence for the mixture with unbiased estimator (Like-SGM and HalfRep-SGM) is similar to the base policy, while converging to better solutions. On the other hand, the learning of GumbelRep-SGM is slower and noisier as its gradient estimation is biased. Figure 11 (Right) shows the results for the hyperparameter setting that has the largest AUC. We can see that the conclusion is very similar to the case using the final performance when comparing the base and the mixture policies under the same type of gradient estimator. However, we can also observe that the gaps between the likelihood-ratio estimators and reparameterization estimators increase.

# E ADDITIONAL EXPERIMENTS

## E.1 EFFECT OF THE NUMBER OF COMPONENTS

We use mixture policies with five components in all experiments in the main text. Here, we study the effect of this choice in classic control environments with unshaped rewards, where differences between the base and the mixture policies are more prominent. We use the same experiment protocol as in Section 5.2 and test mixture policies with two and eight components using the half reparameterization gradient estimator. Specifically, we sweep the same hyperparameters for each variant of mixture policies and rerun the best hyperparameter setting for plotting the learning curves.

Figure 12 shows the learning curves of the best hyperparameter setting and the sensitivity to the entropy scale. We can see that while the results are noisy and not quite consistent across environments, mixture policies with various numbers of components generally outperform the base policy. Note that though the learning curve of the best hyperparameter setting of HalfRep-SGM with $m = 8$ is worse than that of Rep-Squashed, the sensitivity curve indicates that HalfRep-SGM with $m = 8$ dominates Rep-Squashed across different $\alpha$. This again suggests that there is significant noise in the evaluation process. Nevertheless, we can see that mixture policies with various numbers of components are similarly effective. More results for mixture policies with different numbers of compoenents can be found in Appendix E.6.

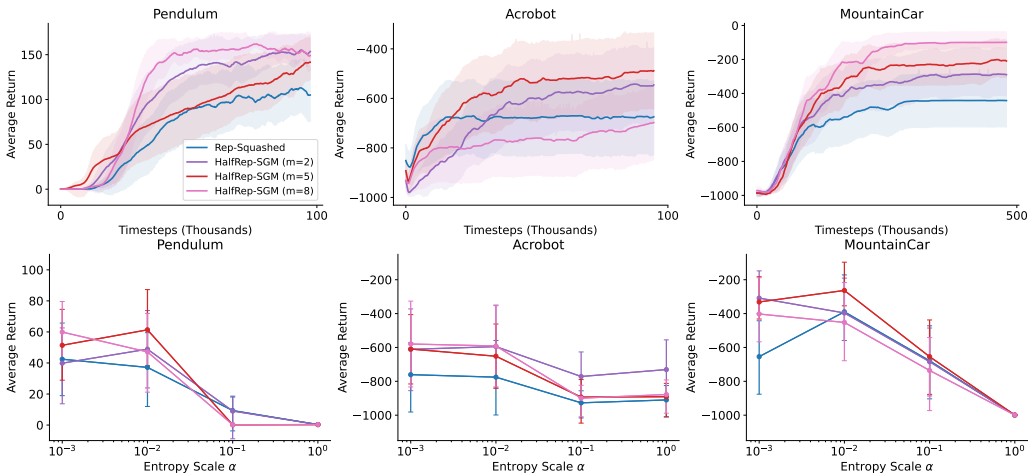

Figure 12: Learning curves and sensitivity curves for mixture policies with different number of components. The shaded area shows the $95\%$ bootstrap confidence intervals across 30 runs. The error bars plot the $95\%$ bootstrap confidence intervals across 10 runs.

## E.2 EFFECT OF USING A FIXED WEIGHTING POLICY

Aside from half reparameterization and Gumbel reparameterization, an alternative way to use reparameterization in mixture policies is by using a fixed weighting scheme, as explored by Baram et al. (2021). However, in general, we lack prior knowledge about what constitutes an effective fixed weighting policy, so a uniform weighting scheme is typically chosen. It is important to note that such a fixed weighting policy can have drawbacks. First, restricting the weighting scheme reduces the flexibility of the policy class, which may be undesirable. Second, mixture policies with fixed weights often require more significant parameter updates when transitioning between distributions.

For instance, when all modes have collapsed to a single mode, it is more challenging for fixed-weight mixture policies to introduce a new mode far from the current mode, as the component locations are constrained near the existing mode due to the non-zero fixed weights. In contrast, mixture policies with learnable weights can focus on a specific mode while keeping other components positioned far away, as their negligible weights minimize their impact on the resulting distribution. In scenarios where the mixture policy needs to introduce a new mode, a learnable-weight policy can simply

adjust the mixing weight of a component that is already near the desired mode, enabling more efficient adaptation.

Following the above intuition, we hypothesize that *mixture policies with a uniform weighting policy will underperform mixture policies with learnable weighting policy.* We test this hypothesis in classic control environments. We use the same experiment protocol as in Section 5.2 and test the uniform-weight mixture policy with reparameterized component policies.

Figure 13 shows the learning curves of the best hyperparameter setting and the sensitivity to the entropy scale. We can see that while the uniform-weight mixture policy (UniformRep-SGM) appears to perform well in Pendulum, it is likely to be worse than HalfRep-SGM and GumbelRep-SGM in the other two environments. Especially in MountainCar, it even seems to underperform the base policy.

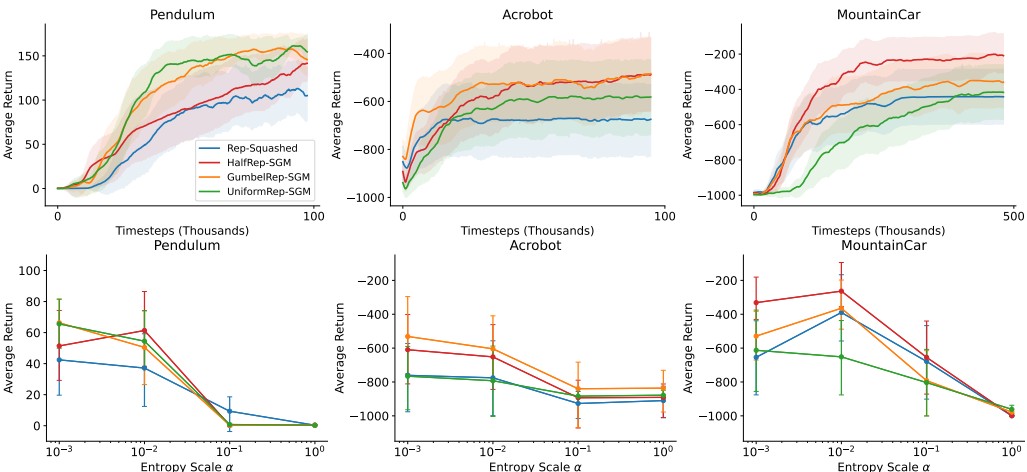

Figure 13: Learning curves and sensitivity curves for different variants of mixture policies. The shaded area shows the $95\%$ bootstrap confidence intervals across 30 runs. The error bars plot the $95\%$ bootstrap confidence intervals across 10 runs.

### E.3  EFFECT OF USING A HEAVY-TAILED BASE POLICY

In principle, the base policy can be any policy and even be different across different components. Here, we consider Cauchy policy Bedi et al. (2024) as the base policy, which is heavy-tailed and promotes persistent exploration. Our hypothesis is that *using mixture policies would also provide benefits when the base policy is Cauchy policy.* We adopt the same experiment protocol as in Section 5.2 and test Cauchy and Cauchy mixture (CM) policies with reparameterization gradient estimators.

Figure 14 shows the learning curves of the best hyperparameter setting and the sensitivity to the entropy scale. In Acrobot and MountainCar, mixture policies appear to be helpful when the base policy is heavy-tailed. However, in Pendulum, Cauchy-based policies all perform significantly worse than the corresponding Gaussian-based policies. This might potentially be due to two reasons: 1) Cauchy-based policies are generally not suitable for this environment, or 2) the preset search range of $\alpha$ is too large for Cauchy-based policies. In summary, we conclude that even when the base policy is heavy-tailed, using mixture policies may still provide benefits in environments with unshaped rewards.

### E.4  RESULTS ON ADDITIONAL ROBOTIC ENVIRONMENTS

In addition to environments from MuJoCo and the DeepMind Control Suite, we conduct experiments on three sparse-reward robotic simulation environments: Pen Rajeswaran et al. (2018), FetchReach Plappert et al. (2018), and FetchSlide Plappert et al. (2018). We use the same experiment protocol as in Section 5.1. Note that, in other more difficult sparse-reward environments like FetchPush and FetchPickAndPlace from Plappert et al. (2018) and ObjectRelocation, DoorOpening, and Hammer

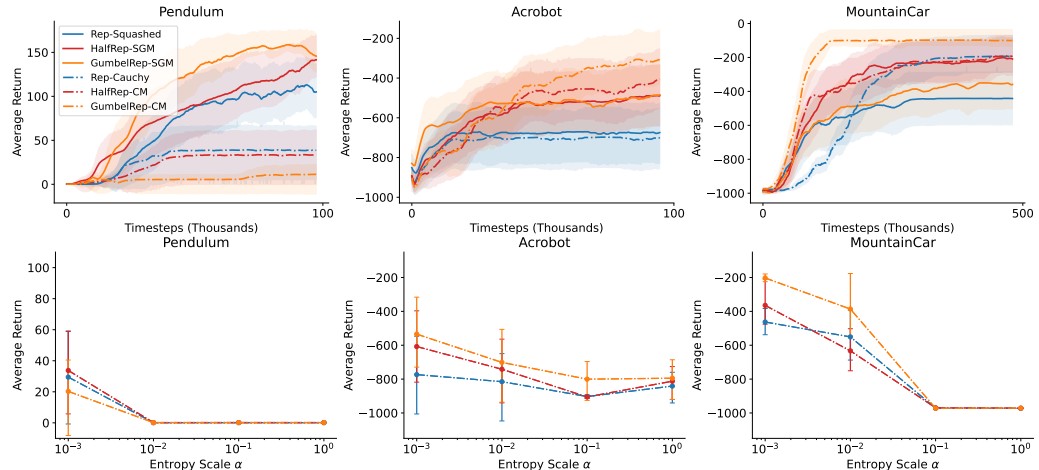

Figure 14: Learning curves and sensitivity curves when the base policy is Cauchy policy. The shaded area shows the 95% bootstrap confidence intervals across 30 runs. The error bars plot the 95% bootstrap confidence intervals across 10 runs.

from Rajeswaran et al. (2018), learning online without any advanced exploration strategy is very difficult as random exploration may seldom or never experience success.

From Figure 15, we can see that the performance difference in Pen and FetchReach is very small. Note that although Pen is introduced with both dense-reward and sparse-reward settings, we found that the returned rewards are also shaped rewards in the latter case despite the magnitude of rewards being much smaller than the former. On the other hand, FetchReach is mentioned to be so easy that even partially broken implementations can learn successfully Plappert et al. (2018). Thus, it is not surprising that we are seeing very minor difference in these two environments. Finally, we observe some performance difference in FetchSlide. While GumbelRep-SGM seems to perform slightly better than the baseline, Rep-Squashed, it is not the case for HalfRep-SGM. It is difficult to draw a conclusion from this limited result, and we leave further investigation for future work.

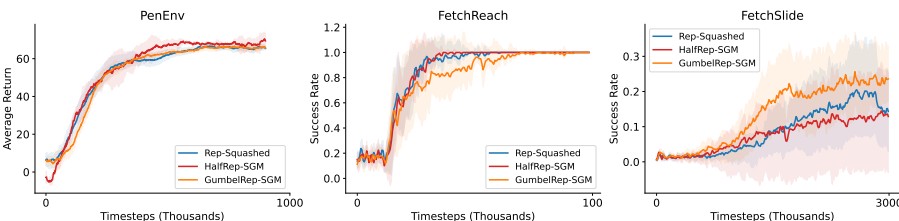

Figure 15: Learning curves in three additional robotic environments. The shaded area shows the 95% bootstrap confidence intervals across 8 runs.

### E.5    MuJoCo Environments with Shifted Rewards

Since we observe more performance improvement in Acrobot and MountainCar, in which the critic is optimistically initialized, we originally hypothesized that mixture policies would exhibit a more obvious performance gain in MuJoCo environments under the same conditions. Thus, we design variants of these environments that have such characteristics.

**Environments.** We use five MuJoCo environments in this experiment: Hopper, Walker2D, HalfCheetah, Ant, and Swimmer. However, we shift the per step reward by $-5$, $-5$, $-15$, $-5$, and $-0.5$, respectively. If the episode terminates before 1000 steps in episodic tasks, including Hopper, Walker2D, and Ant, we assume the agent enters an absorbing state until timestep 1000.

Figure 16 shows the results for this setting. We can see that the performances of the base policy and the mixture variants are also quite close in this setting, invalidating our hypothesis above. Thus, we turned into investigating the separation between environments with shaped rewards versus those with unshaped rewards as in Section 5.2.

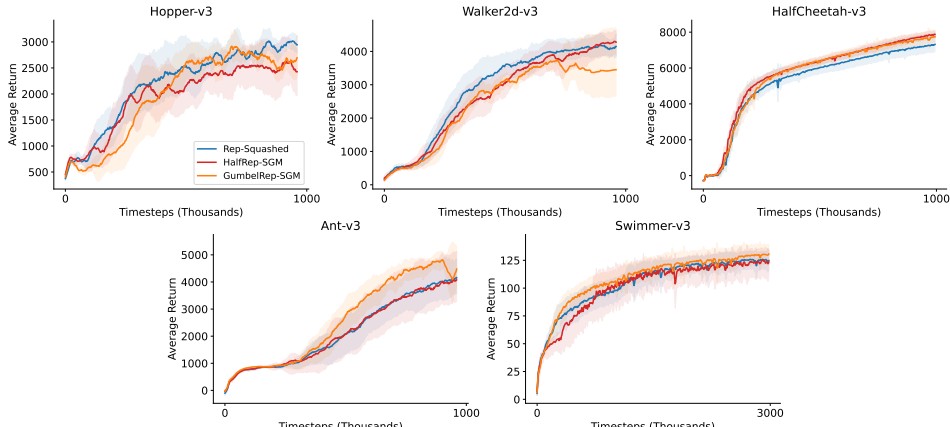

Figure 16: Learning curves in five MuJoCo environments with shifted rewards. The shaded area shows the $95\%$ bootstrap confidence intervals across 8 runs. The shifted rewards are removed in the plot for comparison with the original environment.

**The effect of the temperature parameter $\tau$.** In Figure 17, we also show the results of a hyper-parameter sensitivity study on the temperature parameter $\tau$. As discussed in Section 4.2, a smaller temperature is less stable, while a larger temperature is more stable.

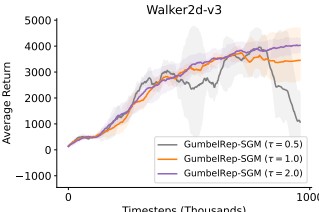

Figure 17: Effect of the temperature parameter $\tau$ in Walker2D with shifted rewards. The shaded area shows the $95\%$ bootstrap confidence intervals across 8 runs. The shifted rewards are removed in the plot for comparison with the original environment.

### E.6 BEYOND SAC AND SQUASHED GAUSSIAN POLICIES

In this section, we present experiments using a different entropy-regularized actor-critic algorithm to further demonstrate the effectiveness of mixture policies and supply an important ablation study.

**Algorithms.** In this experiment, we use Greedy Actor-Critic (GreedyAC; Neumann et al., 2022). GreedyAC updates the policy using the idea of cross-entropy method (Rubinstein, 1999), which is significantly different from SAC. Further, GreedyAC uses the Gaussian policy instead of the squashed Gaussian policy and thus serves as a very good supplement to our study. In addition, GreedyAC does not have a reparameterization gradient estimator but selectively optimizes the log likelihood of good actions. For more details, please refer to Neumann et al. (2022).

**Environments.** We use Pendulum and MountainCar from Neumann et al. (2022). Both environments use a discount factor of $0.99$ and have an episode cut-off of $1000$. Since their version of Pendulum is different from the one we use, we refer to it as Pendulum-v2.

**Experimental details.** We use a two-layer feedforward network with a hidden dimension of $64$, a replay buffer size of $100,000$, and a batch size of $32$. We sweep the initial critic step size $\eta_{q,0} = 10^x$ for $x \in \{-5,-4,-3,-2,-1\}$, the initial actor step size $\eta_{p,0} = \kappa\eta_{q,0}$ for $\kappa \in \{10^{-3}, 10^{-2}, 10^{-1}, 1, 2, 10\}$, and the entropy scale $\alpha = 10^y$ for $y \in \{-3,-2,-1,0,1\}$. Both the actor policy and the proposal policy in GreedyAC use the same initial step size. We run each hyperparameter setting for 10 runs and directly report the performance of the best setting that has the largest area under the learning curve (AUC).

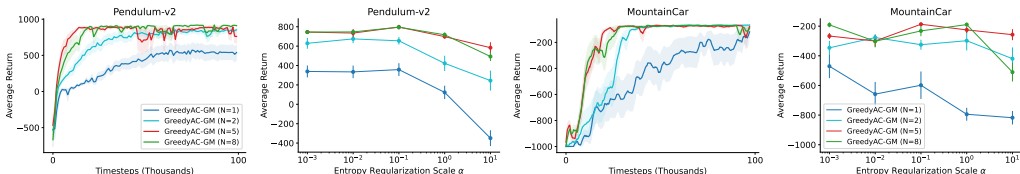

Figure 18: Learning curves and sensitivity curves to $\alpha$ of GreedyAC in Pendulum-v2 and MountainCar. The shaded area and error bars show the standard errors across 10 runs.

**Do mixture policies work in another algorithm with a different policy parameterizations?** Figure 18 shows both learning curves and sensitivity curves to $\alpha$. By comparing GreedyAC-GM (N=1) and GreedyAC-GM (N=5) in both environments, we can see that the mixture policy both helps with learning and is more robust to the entropy scale. Since MountainCar is an environment with sparse signals, the results also suggest that the mixture policy helps with exploration in this case.

**How sensitive are mixture policies to the number of components?** Figure 18 also plots the performance of GreedyAC-GM (N=2) and GreedyAC-GM (N=8). We can see that mixture policies may perform a bit worse with just two components, which may be due to the insufficient flexibility of a two-mixture. However, increasing the number of components from $5$ to $8$ does not show a significant difference. Thus, we hypothesize that the marginal benefit of increasing the number of components will decrease, and using a mixture of a few components should be sufficient in most cases.

# F  PSEUDOCODE

---

**Algorithm 1** Soft Actor-Critic

---

Initialize parameters $\mathbf{w}_1, \mathbf{w}_2, \boldsymbol{\theta}, \bar{\mathbf{w}}_1 \leftarrow \mathbf{w}_1, \bar{\mathbf{w}}_2 \leftarrow \mathbf{w}_2$, replay buffer $\mathcal{B}$

Obtain initial state $S_0$

**while** agent interacting with the environment **do**

Sample action $A_t \sim \pi_{\boldsymbol{\theta}}(\cdot|S_t)$

Take action $A_t$, observe $R_{t+1}, S_{t+1}$

Add $\langle S_t, A_t, S_{t+1}, R_{t+1} \rangle$ to the buffer $\mathcal{B}$

Grab a random mini-batch $B$ from buffer $\mathcal{B}$

Sample $A' \sim \pi_{\boldsymbol{\theta}}(\cdot|S')$ for each transition $\langle S, A, S', R \rangle$ in $B$

Update $\mathbf{w}_i$ on $B$ for $i \in \{1, 2\}$ using

$$\mathbf{w}_i \leftarrow \mathbf{w}_i + \alpha_{q,t} \left( R + \gamma \left( \min_{j \in \{1,2\}} Q_{\bar{\mathbf{w}}_j}(S', A') - \alpha \log \pi_{\boldsymbol{\theta}}(A'|S') \right) - Q_{\mathbf{w}_i}(S, A) \right) \nabla Q_{\mathbf{w}_i}(S, A)$$

Sample $\tilde{A} \sim \pi_{\boldsymbol{\theta}}(\cdot|S)$ for each transition $\langle S, A, S', R \rangle$ in $B$

**if** using the reparameterization estimator **then**

Update $\boldsymbol{\theta}$ on $B$ using

$$\boldsymbol{\theta} \leftarrow \boldsymbol{\theta} + \alpha_{p,t} \left( \nabla_{\boldsymbol{\theta}} \alpha \log \pi_{\boldsymbol{\theta}}(\tilde{A}|S) - (\nabla_{\tilde{A}} \min_{j \in \{1,2\}} Q_{\boldsymbol{\theta}_j}(S, \tilde{A}) - \nabla_{\tilde{A}} \alpha \log \pi_{\mathbf{w}}(\tilde{A}|S)) \nabla_{\mathbf{w}} f_{\mathbf{w}}(\epsilon; S) \right)$$

**else**

Sample $N$ actions $\{A_i\}_{i=1}^N$ from $\pi_{\boldsymbol{\theta}}(\cdot|S)$ and compute baseline $V_b(S) = \frac{1}{N} \sum_{i=1}^N \min_{j \in \{1,2\}} Q_{\boldsymbol{\theta}_j}(S, A_i)$

Update $\boldsymbol{\theta}$ on $B$ using

$$\boldsymbol{\theta} \leftarrow \boldsymbol{\theta} + \alpha_{p,t} \left( \min_{j \in \{1,2\}} Q_{\boldsymbol{\theta}_j}(S, \tilde{A}) - V_b(S) - \alpha \log \pi_{\mathbf{w}}(\tilde{A}|S) \right) \nabla_{\mathbf{w}} \log \pi_{\mathbf{w}}(\tilde{A}|S)$$

**end if**

Update target network weights $\bar{\mathbf{w}}_i$

$$\bar{\mathbf{w}}_i \leftarrow \tau \mathbf{w}_i + (1 - \tau) \bar{\mathbf{w}}_i \text{ for } i \in \{1, 2\}$$

**end while**

---

