# OpenReview forum: "Investigating Mixture Policies in Entropy-Regularized Actor-Critic"
_ICLR.cc/2025/Conference — Submitted to ICLR 2025_

### Official Review · Reviewer_TfHJ · 2024-10-30

**Soundness:** 2
**Presentation:** 2
**Contribution:** 3
**Rating:** 5
**Confidence:** 4

**Summary:**

This paper explores the role of mixture policies in entropy-regularized reinforcement learning. The authors argue that mixture policies can provide better solution quality and robustness compared to standard Gaussian policies. The study aims to bridge the gap in existing algorithms, specifically addressing the challenges associated with the reparameterization of mixture policies.

**Strengths:**

1. The exploration of mixture policies introduces a novel perspective in reinforcement learning, potentially enhancing performance.
2. The theoretical framework is robust, providing a solid basis for the claims made regarding performance improvements.

**Weaknesses:**

1. Lack of comprehensive comparisons with other advanced algorithms that also utilize multimodal policies, e.g. diffusion models that mentioned by the author in the third paragraph of introduction. These models must be considered as baselines in the experiments because the proposed mixture policy has the same aim with them, i.e. multimodal ability.
2. The algorithm seems to be effective only in shaped-reward environments, which are constructed by the authors on purpose. The effectiveness of algorithm should be tested in general environments, rather than designing a special environment tailored specifically for the algorithm after it has been proposed.
3. The computation time efficiency is not elaborated. It is important for performance to be high within acceptable time efficiency. The paper should include comparisons of network forward time, backward time, and the training wall time.
4. Lack the analysis for the number of mixture policies, i.e. $N$. In complex environments, this hyperparameter may be vital. The sensitivity analysis should be implemented.

**Questions:**

1. How is it compared to advanced algorithms that also utilize multimodal policies, e.g. diffusion models?
2. Are there general scenarios or tasks (not shaped-reward envs) where mixture policies significantly outperform Gaussian policies?
3. How about the computation time efficiency of the proposed mixture policy?
4. Does the hyperparameter $N$ affect a lot?

---

> ### Author Response · Authors · 2024-11-21
> **Thank you for your review - I**
>
> Thanks for acknowledging our investigation on mixture policies and for the time you put into reviewing and providing valuable feedback. Here are our responses to each point you raised:
>
> **Q1 & Weakness 1**: How is it compared to advanced algorithms that also utilize multimodal policies, e.g. diffusion models? & Lack of comprehensive comparisons with other advanced algorithms that also utilize multimodal policies, e.g. diffusion models that mentioned by the author in the third paragraph of introduction.
>
> **Response**: Our research focuses on comparing a simple distribution with its mixture distribution as the action distribution, minimizing other differences to isolate the impact of a more flexible policy class. While advanced algorithms may achieve greater performance improvements, our objective is not to benchmark mixture policies against every possible policy class but to provide a focused analysis. Nonetheless, our findings may offer insights into the performance gains observed in more advanced approaches. It is important to note that the performance improvements in these advanced solutions (Tang & Agrawal, 2018; Mazoure et al., 2020; Messaoud et al., 2024; Chao et al., 2024) cannot be solely attributed to the increased flexibility of their policy classes. These approaches often involve multiple changes, such as novel algorithmic designs, larger or more sophisticated networks, and different optimization techniques. Our experimental results in environments with shaped rewards suggest that the observed performance gains in such cases may stem more from algorithmic or optimization enhancements rather than the additional flexibility introduced by the policy class.
>
> Tang, Y., & Agrawal, S. (2018). Implicit policy for reinforcement learning. arXiv preprint arXiv:1806.06798.
>
> Mazoure, B., Doan, T., Durand, A., Pineau, J., & Hjelm, R. D. (2020). Leveraging exploration in off-policy algorithms via normalizing flows. CoRL.
>
> Messaoud, S., Mokeddem, B., Xue, Z., Pang, L., An, B., Chen, H., & Chawla, S. (2024) S2AC: Energy-Based Reinforcement Learning with Stein Soft Actor Critic. ICLR.
>
> Chao, C. H., Feng, C., Sun, W. F., Lee, C. K., See, S., & Lee, C. Y. (2024). Maximum Entropy Reinforcement Learning via Energy-Based Normalizing Flow. NeurIPS.
>
> **Q2 & Weakness 2**: Are there general scenarios or tasks (not shaped-reward envs) where mixture policies significantly outperform Gaussian policies? & The algorithm seems to be effective only in shaped-reward environments, which are constructed by the authors on purpose. The effectiveness of algorithm should be tested in general environments, rather than designing a special environment tailored specifically for the algorithm after it has been proposed.
>
> **Response**: To clarify, our observation is that mixture policies help in environments with unshaped rewards instead of shaped rewards. We would also like to point out that Acrobot and MountainCar are well-known classic control environments with unshaped rewards that we do not “tailor”. As suggested by Reviewer c4Ff, we additionally tested three sparse reward environments (two of which have unshaped rewards). The results are presented in Appendix E.4 in the updated submission. In summary, the performance difference in two of the environments is quite small as they are quite easy. In the remaining environment, one variant of mixture policies (GumbelRep-SGM) performs slightly better than the baseline, while the other variant performs slightly worse (potentially due to the high variance of the likelihood ratio part of the half-reparameterization estimator). While it’s difficult to draw a conclusion from this limited result, we leave further investigation for future work.
>
> On the other hand, the observation that they don’t help when the rewards are shaped is reasonable, as the need for exploration is alleviated by the guided rewards. This observation might also provide insights into other works in the literature that study complex policy classes, which is discussed in our response to Q1.

---

> > ### Author Response · Authors · 2024-11-21
> > **Thank you for your review - II**
> >
> > **Q3 & Weakness 3**: How about the computation time efficiency of the proposed mixture policy? & The computation time efficiency is not elaborated. It is important for performance to be high within acceptable time efficiency. The paper should include comparisons of network forward time, backward time, and the training wall time.
> >
> > **Response**: The computation time for training mixture policies depends on the type of the gradient estimator. In general, when both the base policy and the mixture policy don’t use a baseline, the additional computation for the mixture policy is smaller since we only need extra heads in the actor network for the mixture parameters. Further, Like-Squashed, Like-SGM, and HalfRep-SGM require a baseline to reduce the variance and stabilize learning, which makes them slower compared to those that don’t need a baseline (Rep-Squashed, GumbelRep-SGM). In our experiment, we don’t use a separate state-value (v-value) network for the baseline but compute it using the average of a batch of $30$ state-action values (q values). We included a table in Appendix D.1 in the updated paper for reference.
> >
> > **Q4 & Weakness 4**: Does the hyperparameter $N$ affect a lot? & Lack the analysis for the number of mixture policies, i.e. $N$. In complex environments, this hyperparameter may be vital. The sensitivity analysis should be implemented.
> >
> > **Response**: We included an investigation of the effect of $N$ in Appendix E.6 in the updated paper (Appendix E.2 in the original submission) using a different entropy-regularized algorithm (GreedyAC) and a different base policy (unsquashed Gaussian). We additionally run the same analysis for SAC with squashed Gaussian as the base policy. The results are presented in Appendix E.1 in the updated paper. In summary, while the results are quite noisy, mixture policies with different $N$ are similarly effective. Note that mixture policies with only two components already improve performance over the base policy.
> >
> > We thank you again for your helpful feedback to improve the paper and hope our response has addressed your concerns. Please let us know if you have further questions.

---

> ### Comment · Area_Chair_f6SA · 2024-11-23
> **From AC.**
>
> Reviewer TfHJ: if possible, can you reply to the rebuttal?

---

### Official Review · Reviewer_u6G7 · 2024-11-03

**Soundness:** 1
**Presentation:** 2
**Contribution:** 2
**Rating:** 5
**Confidence:** 4

**Summary:**

This paper investigates the use of mixture policies in entropy-regularized reinforcement learning, highlighting their flexibility compared to base policies like Gaussians. The authors theoretically demonstrate that mixture policies can improve solution quality and robustness to entropy scaling. They address the challenge of reparameterization for lower variance gradient updates, introducing novel reparameterization gradient estimators for mixture policies. Extensive experiments across various environments reveal that mixture policies enhance exploration efficiency in tasks with unshaped rewards while maintaining comparable performance to base policies in shaped reward scenarios, and exhibit greater robustness to multimodal critic surfaces.

**Strengths:**

1. The paper introduces reparameterization gradient estimators for entropy-regularized reinforcement learning.
2. Theoretical analysis demonstrates that the Gaussian mixture model (GMM) policy outperforms the base Gaussian policy.

**Weaknesses:**

1. The paper lacks motivation regarding the necessity of the reparameterization gradient estimator.
2. Some analyses are trivial and offer limited insights (please see questions）

**Questions:**

1. The motivation for this work is insufficient. Given that the basic likelihood ratio gradient estimator can be applied in the GMM context, what is the rationale for developing the reparameterization gradient estimator? If the aim is variance reduction, the authors should emphasize the theoretical improvements and provide empirical evidence supporting this reduction compared to other estimators like the likelihood ratio estimator, which is currently lacking in the paper.
2. The authors should clarify the novelty of their work. Is the proposed reparameterization gradient estimator used in other reinforcement learning contexts beyond entropy-regularized RL?
3. In Section 3, the authors assert that "the mixture policy is at least as good as or better than the base policy." While the claim of "at least as good" is theoretically supported, how is the assertion of "better" supported theoretically?
4. The authors mention that Gumbel reparameterization introduces bias. Is there any analysis on the scale and impact of this bias on gradient estimation?
5. In Section 6, line 510, the authors state that "the primary limitation is that we used default hyperparameters for the mixture policies in MuJoCo; such defaults were tuned for SAC with the base policy." The authors should discuss additional limitations regarding their theoretical analysis and experimental settings.

---

> ### Author Response · Authors · 2024-11-21
> **Thank you for your review**
>
> Thank you for your time put into reviewing and providing valuable feedback. Here are our responses to each point you raised:
>
> **Q1 & Weakness 1**: The motivation for this work is insufficient. Given that the basic likelihood ratio gradient estimator can be applied in the GMM context, what is the rationale for developing the reparameterization gradient estimator?
>
> **Response**: The purpose of the derived reparameterization (RP) estimators is to fill in the gap in the literature that no such estimators existed prior to our work. We included a thorough discussion of relevant works we found in Appendix A. In principle, RP estimators are not necessary for subsequent use in SAC. However, in the v1 version of Haarnoja et al. (2018b, https://arxiv.org/pdf/1801.01290v1), SAC is shown to perform more stable with the RP gradient. Despite the RP gradient (including deterministic gradient) being considered to have lower variance and used in many important baselines (DDPG, TD3, SAC, etc.), there aren't many known results about their superiority over the likelihood ratio (LR) estimator (see [1] for a more comprehensive discussion). While we leave the open question of the comparison between the PR and LR estimators for future work, we consider the derivation of PR estimators for mixture policies and show their effectiveness as a contribution to the community.
>
> [1] Parmas, P., & Sugiyama, M. (2021). A unified view of likelihood ratio and reparameterization gradients. Aistats.
>
> **Q2**: The authors should clarify the novelty of their work. Is the proposed reparameterization gradient estimator used in other reinforcement learning contexts beyond entropy-regularized RL?
>
> **Response**: We discuss relevant work in Appendix A and clarify our contributions at the end of that section. Beyond the most closely related works, we also identify two studies that employ a gradient estimator similar to the Gumbel reparameterization estimator: one in the discrete action setting [2] and another in the multi-agent setting [3]. To the best of our knowledge, our proposed reparameterization gradient estimators for mixture policies are novel. Importantly, the novelty of our work extends beyond the development of these gradient estimators. Specifically, our contributions include:
> 1. Providing new insights into the impact of a more flexible policy class on optimality in entropy-regularized reinforcement learning settings.
> 2. Investigating the advantages of using mixture policies in environments with unshaped rewards, along with offering a deeper understanding of their behavior in such scenarios.
> This combination of theoretical and empirical contributions advances the understanding of mixture policies in reinforcement learning.
>
> [2] Fan, T. H., & Wang, Y. (2022). Soft actor-critic with integer actions. ACC.
>
> [3] Tilbury, C. R., Christianos, F., & Albrecht, S. V. (2023). Revisiting the Gumbel-Softmax in MADDPG. arXiv.
>
> **Q3 & Weakness 2**: how is the assertion of "better" supported theoretically?
>
> **Response**: It’s supported by the counterexample (shown in Figure 1). In this counterexample, the mixture policy still has stationary points even when the base policy does not (see $\alpha=0.4$ in Figure 1 Middle). We will turn this counterexample into a proposition statement in future versions of our paper.
>
> **Q4**: Is there any analysis on the scale and impact of this bias on gradient estimation (of Gumbel reparameterization)?
>
> **Response**: We included an experiment on this bias-variance trade-off in Appendix E.5 in the updated paper (Figure 17). Specifically, we tested three different temperatures of Gumbel-softmax in a variant of Walker2d. We observe that while $\tau=2.0$ performs similarly to $\tau=1.0$, $\tau=0.5$ suffers from instability due to larger variance. In addition, we observe that the bias of using $\tau=1.0$ is more salient in the bandit setting (Fig 6. Right, comparing HalfRep-SGM and GumbelRep-SGM) but not in the MDP setting (Fig 2 and Fig 4). Our hypothesis is that in the MDP setting, the noise and bias of the critic during training dominate the bias of Gumbel-softmax.
>
> **Q5**: The authors should discuss additional limitations regarding their theoretical analysis and experimental settings.
>
> **Response**: We thank all reviewers for pointing out additional limitations. While we have provided additional experiments on an extra baseline and a few new environments in the updated version, we will also incorporate the following discussion on the remaining limitations of our work: 1) Our paper does not provide a theoretical analysis of the comparison between likelihood ratio estimator and the reparameterization estimator, which is an interesting open question. 2) Our experiments only cover environments for robotics manipulation and classic control. Investigation on more continuous control domains is left for future work.
>
> We hope our response has addressed your concerns. Please let us know if you have further questions.

---

> > ### Comment · Reviewer_u6G7 · 2024-11-26
> > **Response to author rebuttal**
> >
> > I appreciate the authors' clarifications and the additional experiments provided. Based on this, I have decided to raise my score.

---

> ### Comment · Area_Chair_f6SA · 2024-11-23
> **From AC.**
>
> Reviewer u6G7: if possible, can you provide a response to the rebuttal?

---

### Official Review · Reviewer_wfLs · 2024-11-03

**Soundness:** 2
**Presentation:** 2
**Contribution:** 2
**Rating:** 3
**Confidence:** 4

**Summary:**

The paper investigates mixture policies in entropy-regularized reinforcement learning, highlighting their advantages over traditional base policies like Gaussians. It provides theoretical insights into their improved solution quality and robustness, while addressing the challenges in implementing these policies effectively in algorithms such as Soft Actor-Critic.

**Strengths:**

a). The paper provides solid theoretical backing for the proposed methods, elucidating their benefits over conventional Gaussian policies.

b). The focus on multimodal entropy-regularized frameworks is timely, considering the current trends in reinforcement learning research.

**Weaknesses:**

a). There is no performance gain in complex environments (such as Hopper, Walker, Ant, Humanoid, etc.), and the effective shaped reward environments only demonstrate improvements in simpler environments (like Pendulum, Acrobot, Mountain), which is not very convincing.

b). The figures in the third row of Fig 4 actually demonstrates hyperparameter sensitivity rather than the robustness described in line 371. Although the return is higher than Rep-Squashed, the larger variation compared to Rep-Squashed when $\alpha$ changes indicates that hyperparameter sensitivity is not as good as that of Rep-Squashed. Additionally, testing for robustness should involve adding noise to the observations.

c). It is not enough to only compare with SAC. The paper should compare with Daniel (2012), Celik (2022), Nematollahi (2022), and Seyde (2022). These algorithms are all mixture policies (the author describes in Appendix A). Also, the paper should compare with Hou (2020), Baram (2021).

**Questions:**

In Figure 1 (left), the GM policy does not completely fit the reward function. What is the reason?

---

> ### Author Response · Authors · 2024-11-21
> **Thank you for your review - I**
>
> Thank you for your effort in reviewing our paper and providing valuable feedback. Here are our responses to your concerns:
>
> **Q1**: In Figure 1 (left), the GM policy does not completely fit the reward function. What is the reason?
>
> **Response**: Note that the density of the GM policy would not completely fit the reward function in general. In the extreme case where the entropy regularization $\alpha$ is $0$, the density would concentrate on the points of optimal actions with zero probability mass elsewhere. In the other extreme case where the entropy regularization $\alpha$ is very large, the density would scatter across the action space. In Figure 1 (left), the optimal GM policy looks quite similar to the reward function because of a moderate $\alpha$.
>
> **Weakness 1**: There is no performance gain in complex environments (such as Hopper, Walker, Ant, Humanoid, etc.), and the effective shaped reward environments only demonstrate improvements in simpler environments (like Pendulum, Acrobot, Mountain), which is not very convincing.
>
> **Response**: The goal of our research is not to assert that mixture policies, as a type of policy parameterization, are inherently superior to base policies. Instead, we aim to understand their consequences and benefits, including the contexts in which they provide advantages. One of our key findings—that mixture policies do not consistently improve performance in commonly used environments with shaped rewards—yields important insights. While much of the literature seeks to motivate the use of multimodal policies, our results suggest that the performance gains reported in such studies may not stem solely from the added flexibility of the policy class. These proposed solutions often incorporate other factors, such as new algorithmic designs, larger or different network architectures, and alternative optimization methods (Tang & Agrawal, 2018; Mazoure et al., 2020; Messaoud et al., 2024; Chao et al., 2024).
>
> Our research isolates the effect of policy class flexibility by comparing a simple distribution to its mixture counterpart as the action distribution while keeping other factors constant. This approach provides clear and focused insights into the impact of having a more flexible policy class.
>
> The implications of our findings are twofold:
> 1. In environments with shaped rewards, a more flexible policy class may not be necessary, and observed improvements in previous studies might instead result from algorithmic or optimization differences.
> 2. In environments with less shaped rewards, mixture policies or other more flexible policy classes are promising approaches that could yield significant benefits.
>
> Tang, Y., & Agrawal, S. (2018). Implicit policy for reinforcement learning. arXiv preprint arXiv:1806.06798.
>
> Mazoure, B., Doan, T., Durand, A., Pineau, J., & Hjelm, R. D. (2020). Leveraging exploration in off-policy algorithms via normalizing flows. CoRL.
>
> Messaoud, S., Mokeddem, B., Xue, Z., Pang, L., An, B., Chen, H., & Chawla, S. (2024) S2AC: Energy-Based Reinforcement Learning with Stein Soft Actor Critic. ICLR.
>
> Chao, C. H., Feng, C., Sun, W. F., Lee, C. K., See, S., & Lee, C. Y. (2024). Maximum Entropy Reinforcement Learning via Energy-Based Normalizing Flow. NeurIPS.

---

> ### Author Response · Authors · 2024-11-21
> **Thank you for your review - II**
>
> **Weakness 2**: The figures in the third row of Fig 4 actually demonstrates hyperparameter sensitivity rather than the robustness described in line 371. Although the return is higher than Rep-Squashed, the larger variation compared to Rep-Squashed when $\alpha$ changes indicates that hyperparameter sensitivity is not as good as that of Rep-Squashed. Additionally, testing for robustness should involve adding noise to the observations.
>
> **Response**: We apologize for the potentially confusing use of the term “robustness”. In our paper, by stating that “the mixture policy is more robust to the entropy scale”, we meant that the mixture policy performs better or at least as well as the base policy across different values of $\alpha$. We will clarify this terminology in future versions of our paper to avoid confusion.
>
> Regarding the concern about hyperparameter sensitivity, we acknowledge that the figures in the third row of Fig. 4 demonstrate a larger variation in performance for the mixture policy compared to Rep-Squashed as $\alpha$ changes, indicating greater sensitivity to this hyperparameter. However, we would like to emphasize that the mixture policy achieves higher returns than Rep-Squashed quite consistently across the tested range of $\alpha$.
>
> Additionally, we recognize that testing for robustness in a broader sense should involve scenarios such as adding noise to observations. We appreciate this valuable feedback and will consider incorporating such tests in future work to provide a more comprehensive evaluation of robustness.
>
> **Weakness 3**: It is not enough to only compare with SAC. The paper should compare with Daniel (2012), Celik (2022), Nematollahi (2022), and Seyde (2022). These algorithms are all mixture policies (the author describes in Appendix A). Also, the paper should compare with Hou (2020), Baram (2021).
>
> **Response**: We focus on treating mixture distribution as a more complex distribution compared to the base distribution instead of proposing a new algorithm with a new policy class. Among the works you mentioned, Baram (2021) is the most relevant to our pursuit, which also studies mixture policies as a more complex policy class. We tested their approach in the classic control domains, and the results are presented in Appendix E.2 in the updated paper. Note that our proposed gradient estimators alleviate SACM’s (denoted as UniformRep-SGM in Figure 13) restriction on mixture policies and perform consistently better, while SACM’s performance can sometimes be worse than the base policy.
>
> We thank you again for your feedback and hope our response has addressed your concerns. Please let us know if you have further questions.

---

> > ### Comment · Reviewer_wfLs · 2024-11-26
> > **Response to author rebuttal**
> >
> > Thank you for the rebuttal response.
> > I understand the goal of our research is not to assert that mixture policies are inherently superior to base policies, but the lack of performance improvement in complex environments, coupled with performance gains in simple environments, seems to be a systemic phenomenon. You should provide more explanation and description; otherwise, this might lead people to believe that your policy is only suitable to toy tasks. The greater sensitivity to hyperparameter may be considered to solve in your future works. For these reasons, I maintain my original score.

---

> ### Comment · Area_Chair_f6SA · 2024-11-23
> **From AC.**
>
> Reviewer wfLs: if possible, can you provide a response to the rebuttal?

---

### Official Review · Reviewer_J7Ed · 2024-11-04

**Soundness:** 3
**Presentation:** 2
**Contribution:** 2
**Rating:** 3
**Confidence:** 4

**Summary:**

The paper (re-)introduces the use of mixture policies into policy gradient methods for entropy-regularized reinforcement learning (RL). The mixture policies considered consist of convex combinations of component policies, where the convex weightings are themselves a parameterized function. The primary component policies considered are Gaussian policies and modifications thereof, and the primary RL method considered is Soft Actor-Critic (SAC). Theoretical results are provided for the bandit setting showing that mixture policies perform at least as well as their component policies and any first-order stationary point of the component policies also corresponds to a stationary point of any associated mixture policy. To enable the variance reduction benefits of reparametrized policy gradients, half- and fully-reparametrized policy gradient theorems are derived, where the half reparametrization reparametrizes only the component policies, while the full reparametrization employs a Gumbel-softmax scheme to reparametrize the weighting function as well. Experimental results on MuJoCo and DeepMind control benchmark tasks are provided that compare SAC using mixture policies with vanilla SAC using non-mixture policies. The results indicate that mixture policies result in improved performance over vanilla SAC on sparse-reward tasks, likely due to improved exploration.

**Strengths:**

The key contributions of the paper include:
1. Mixture policies are reintroduced, primarily for use with SAC. Note that mixture policies were previously introduced (see related works in appendix), but not extensively studied.
2. The experimental results suggest that mixture policies lead to improved exploration in sparse-reward settings.
3. The theoretical results for the bandit setting provide some reassurance as to the usefulness of mixture policies.

**Weaknesses:**

There paper suffers from several drawbacks:
1. The experimental results are limited and inconclusive. While mixture policies are shown to lead to improved ending performance on some (not all) of the sparse-reward (I understood "shaped" as referring to dense rewards and "unshaped" as referring to sparse rewards) tasks considered, the performance improvement is marginal on most problems. Though this indicates that mixture policies are potentially promising and worth further investigation, the results are suggestive rather than conclusive. More importantly, no comparisons of the proposed mixture policies with comparable exploration-oriented methods are provided, despite the existence of many such methods in the literature (e.g., [Kobayashi, 2019] and [Bedi et al., 2024] mentioned in the introduction). Without such comparisons, it is unclear how the contribution of this work fits into the broader literature.
2. The main theoretical results of Sec. 3 are for simple bandit settings, seriously limiting the significance of the results provided. The proofs appear to be straightforward (I skimmed but did not verify in detail the appendix), containing no major technical innovation to bolster the significance of the analysis. In addition, some Sec. 3 results are not particularly helpful: Prop. 3.2 provides bounds for entropy-constrained problems (not considered elsewhere in the paper) without relating these back to the entropy-regularized problems under consideration; Prop. 3.3 shows that a simple Gaussian policy may fail to have a stationary point, while Prop. 3.4 shows that any stationary point for all component policies is also a stationary point for all corresponding mixture policies -- this does not show that mixture policies have stationary points even when component policies do not, however, which is what we appear to be promised at the beginning of Sec. 3.2. This confusion weakens the theoretical contribution of the paper.
3. The reparametrization policy gradient results of Sec. 4 are poorly motivated: variance reduction is the primary motivation for the use of reparametrized policy gradients, yet the new reparametrizations provided lack variance reduction guarantees, as admitted at the beginning of Sec. 4.1. This raises the question: why should we do reparametrization at all? The lack of clarity on this point detracts from the overall motivation of the approach.
4. Mixture policies are not new. The third paragraph of the introduction as well as the appendix provide a number of references considering them. It is hypothesized in the introduction that "lack of reparametrization estimators for mixture policies" might be the reason mixture policies have not gained more widespread use, but it is left unclear what challenges the lack of such estimators present. Overall, the paper seeks to join the existing literature on mixture policies, but it is unclear how the paper improves over that literature. Yes, reparametrization estimators are provided, but it is unclear why they are needed (and also see issue 3 above).

**Questions:**

1. Have you compared the use of mixture policies with other exploration-inducing policies (e.g., heavy-tailed policies)? If not, why not?
2. What are the main technical innovations that were necessary to performing the analysis?
3. How does Prop. 3.2 relate to the rest of the paper?
4. How do Prop. 3.3 and 3.4 together demonstrate that mixture policies "may have stationary points in scenarios where the base policy does not"?
5. What is the primary purpose of the reparametrization results of Sec. 4? Are they necessary for subsequent use in SAC? Are no variance bounds possible?

---

> ### Author Response · Authors · 2024-11-21
> **Thank you for your review - I**
>
> Thank you for your effort in reviewing our paper and providing valuable comments. We appreciate your acknowledgment of our contributions. Here are our responses to your concerns:
>
> **Q1**: Have you compared the use of mixture policies with other exploration-inducing policies (e.g., heavy-tailed policies)? If not, why not?
>
> **Response**: We didn’t compare the use of mixture policies with other exploration-inducing policies. We chose to focus on our experiment setting because the choice of having a mixture is orthogonal to the choice of the base policy. To demonstrate that mixture policies can also provide benefits when the base policy is heavy-tailed. We conducted additional experiments using the Cauchy policy (Bedi et al., 2024) as the base policy. The results are presented in Appendix E.3 in the updated paper. We found that mixture policies do provide benefits in this case.
>
> **Q2**: What are the main technical innovations that were necessary to performing the analysis?
>
> **Response**: We don’t claim that the techniques in our analysis involve novel treatments. However, despite the analysis being straightforward, such results have not been discussed or made explicit in the literature. We hope our paper provides a ground for future discussion on the properties of more complex policy classes by providing these results.
>
> **Q3**: How does Prop. 3.2 relate to the rest of the paper?
>
> **Response**: We appreciate the reviewer for bringing up this question. Prop. 3.2 operates under a different objective than the standard MaxEnt objective. However, the constrained form is not unfamiliar in MaxEnt RL. In fact, it is the objective of SAC with automatic entropy tuning (Haarnoja et al., 2018c). Thus, Prop. 3.2 makes it clear that mixture policies also have a better optimal stationary point compared to the base policy in that regime.
>
> **Q4**: How do Prop. 3.3 and 3.4 together demonstrate that mixture policies "may have stationary points in scenarios where the base policy does not"?
>
> **Response**: Prop. 3.3 and 3.4 together don’t demonstrate this point but that the mixture policy will also have stationary points when the base policy does. To demonstrate the point you mentioned, we used a counterexample (shown in Figure 1). In this counterexample, the mixture policy still has stationary points even when the base policy does not (see $\alpha=0.4$ in Figure 1 Middle).
>
> **Q5 & Weakness 3**: What is the primary purpose of the reparametrization results of Sec. 4? Are they necessary for subsequent use in SAC? Are no variance bounds possible? … why should we do reparametrization at all? The lack of clarity on this point detracts from the overall motivation of the approach.
>
> **Response**: The purpose of the derived reparameterization (RP) estimators is to fill in the gap in the literature that no such estimators existed prior to our work. We included a thorough discussion of relevant works we found in Appendix A. In principle, reparameterization estimators are not necessary for subsequent use in SAC. However, in the v1 version of Haarnoja et al. (2018b, https://arxiv.org/pdf/1801.01290v1), SAC is shown to perform more stable with the RP gradient. Despite the RP gradient (including deterministic gradient) is considered to have lower variance and used in many important baselines (DDPG (Lillicrap et al., 2016), TD3 (Fujimoto et al., 2018), SAC, etc.), there aren't many known results about their superiority over the likelihood ratio (LR) estimator (see Paramas and Sugiyama (2021) for a more comprehensive discussion). While we leave the open question of the comparison between the PR and LR estimators for future work, we consider the derivation of PR estimators for mixture policies a contribution to the community.
>
> Lillicrap, T. P., Hunt, J. J., Pritzel, A., Heess, N., Erez, T., Tassa, Y., ... & Wierstra, D. (2016). Continuous control with deep reinforcement learning. ICLR.
>
> Fujimoto, S., Hoof, H., & Meger, D. (2018). Addressing function approximation error in actor-critic methods. ICML.
>
> Parmas, P., & Sugiyama, M. (2021). A unified view of likelihood ratio and reparameterization gradients. Aistats.

---

> ### Author Response · Authors · 2024-11-21
> **Thank you for your review - II**
>
> **Weakness 1-1**: While mixture policies are shown to lead to improved ending performance on some (not all) of the sparse-reward (I understood "shaped" as referring to dense rewards and "unshaped" as referring to sparse rewards) tasks considered, the performance improvement is marginal on most problems. Though this indicates that mixture policies are potentially promising and worth further investigation, the results are suggestive rather than conclusive.
>
> **Response**: We deliberately use the terms “shaped/unshaped” instead of “dense/sparse” to avoid potential confusion regarding “-1” per-step rewards (e.g., a reward of -1 at each step until task completion). While such rewards are technically unshaped, some researchers might interpret them as dense rather than sparse.
>
> Excluding “too-easy” unshaped reward tasks where the base policy already performs exceptionally well (e.g., cartpole: balance_sparse, ball_in_cup: catch) and “too-difficult” tasks where reaching the goal state is highly improbable (e.g., acrobot: swingup_sparse in Fig. 8), we observed consistent improvements with mixture policies over the base policy. This includes tasks like cartpole: swingup_sparse and other classical control tasks (see Fig. 4), across gradient estimators (RP and LR estimators in Fig. 4), algorithms, and parameterizations (see Fig. 18 in App. E.6 in the updated paper for results using a different algorithm with an unsquashed Gaussian policy).
>
> These results demonstrate the robustness and versatility of mixture policies, supporting our claim that the findings are both significant and conclusive to a meaningful extent.
>
> **Weakness 1-2**: More importantly, no comparisons of the proposed mixture policies with comparable exploration-oriented methods are provided, despite the existence of many such methods in the literature (e.g., [Kobayashi, 2019] and [Bedi et al., 2024] mentioned in the introduction). Without such comparisons, it is unclear how the contribution of this work fits into the broader literature.
>
> **Response**: We addressed this concern in our response to Q1.
>
> **Weakness 2-1**: The main theoretical results of Sec. 3 are for simple bandit settings, seriously limiting the significance of the results provided.
>
> **Response**: Note that Prop 3.1 and Prop 3.2 in Sec. 3.1 apply to the MDP setting. In addition, Prop 3.4 can also be extended to the MDP setting, which we can include in future versions of our paper. As for Prop 3.3, it is meant to *demonstrate* the point that the base policy will lose its stationary point with large $\alpha$s; an intuitive counterexample, even in the bandit case, would suffice.
>
> **Weakness 2-2**: The proofs appear to be straightforward (I skimmed but did not verify in detail the appendix), containing no major technical innovation to bolster the significance of the analysis. In addition, some Sec. 3 results are not particularly helpful: Prop. 3.2 provides bounds for entropy-constrained problems (not considered elsewhere in the paper) without relating these back to the entropy-regularized problems under consideration; Prop. 3.3 shows that a simple Gaussian policy may fail to have a stationary point, while Prop. 3.4 shows that any stationary point for all component policies is also a stationary point for all corresponding mixture policies -- this does not show that mixture policies have stationary points even when component policies do not, however, which is what we appear to be promised at the beginning of Sec. 3.2. This confusion weakens the theoretical contribution of the paper.
>
> **Response**: We addressed this concern in our response to Q2 to Q4.
>
> **Weakness 4**: It is hypothesized in the introduction that "lack of reparametrization estimators for mixture policies" might be the reason mixture policies have not gained more widespread use, but it is left unclear what challenges the lack of such estimators presents. Overall, the paper seeks to join the existing literature on mixture policies, but it is unclear how the paper improves over that literature.
>
> **Response**: Note that we hypothesized the “lack of reparameterization estimators for mixture policies might be the reason that later versions of SAC switched to a single Gaussian (with reparameterization estimator)” but not why they have not gained more widespread use. In fact, our empirical investigation found that mixture policies don’t improve performance in commonly used MuJoCo environments, which all have shaped rewards. One can hypothesize that the concentration on this limited benchmark might be the reason why mixture policies have not gained more attention. In our work, we demonstrated the difference in mixtures’ effectiveness between environments with shaped and unshaped rewards, which establishes a step toward understanding the effect of policy parameterization on a wider variety of environments.
>
> We hope our responses address your concerns. Please let us know if your concerns remain or if you have other questions.

---

> > ### Comment · Reviewer_J7Ed · 2024-11-25
> > **Response to author rebuttal**
> >
> > Thanks to the authors for their response. First, I appreciate the additional experiments presented in E.3, and they have partially addressed my concern expressed in Weakness 1 and Question 2 of my review about lack of comparison with existing exploration-inducing methods. I also understand that Propositions 3.1 and 3.2 apply to MDPs, as the author mention in their response to Weakness 2-1. Nonetheless, my primary concerns remain: (1) the experimental results are suggestive rather than conclusive, (2) the theoretical analysis contains little technical novelty (see author response to Q2) and the results are limited, (3) the importance of reparametrization policy gradients is insufficiently motivated, and (4) given that mixture policies have already been studied in the literature and the benefits of reparametrization are unsubstantiated, it remains unclear how the present work improves over the existing literature. For these reasons I maintain my original assessment.

---

> ### Comment · Area_Chair_f6SA · 2024-11-23
> **From AC.**
>
> Reviewer J7Ed: if possible, can you provide a response to the rebuttal?

---

### Official Review · Reviewer_c4Ff · 2024-11-09

**Soundness:** 3
**Presentation:** 3
**Contribution:** 3
**Rating:** 5
**Confidence:** 4

**Summary:**

This paper investigates the use of mixture policies in entropy-regularized actor-critic algorithms for reinforcement learning in continuous action spaces. The authors present theoretical properties of mixture policies, showing they have a higher regularized objective and a higher unregularized objective if an entropy-constrained optimization is used as compared to base unimodal policies (single Gaussian policies). They derive two reparameterization gradient estimators for mixture policies: a partial "half-reparameterization" with unbiased gradients and a full reparameterization with biased gradients using the Gumbel-softmax trick. Experiments are conducted on MuJoCo, DeepMind Control Suite, OpenAI classic control environments, and continuous bandits.

**Strengths:**

1. The paper introduces novel reparameterization gradient estimators for mixture policies, filling a gap in the literature. Prior work [1] did not utilize a learnable weighting policy to combine the mixture components that this paper addresses.
2. It improves on existing work, SACM [1], that also investigated mixture policies and updated them using stochastic gradients. While SACM does not report any clear benefits of using it over base policies (SAC), this paper provides claims and evidence of their benefits in sparse reward settings.
3. Theoretical foundations are well-established, providing a solid basis for the proposed methods.
4. Experimental validation across diverse environments supports claims that mixture policies are comparable to unimodal policies in terms of performance.
5. The paper is well-structured, with clear explanations of the motivation, methods, and results.
6. Visualizations (Figures 1 and 5) effectively illustrate key points and findings.

[1] Baram, Nir, Guy Tennenholtz, and Shie Mannor. "Maximum entropy reinforcement learning with mixture policies." arXiv preprint arXiv:2103.10176 (2021).

**Weaknesses:**

1. The sparse reward environments investigated in the paper are simplistic. ‘Pendulum’, ‘MountainCar’, and ‘Acrobot’ are generally treated as toy domains. Experiments on more realistic sparse-reward domains such as ‘Fetch’ [2] and/or ‘Adroit’ [3] domains are missing. This limits the applicability and scalability of the proposed methods.
2. Comparison of the proposed methods against existing work using mixture policies such as SACM [1] in Section 5.
3. A discussion on the benefits of using mixture policies as opposed to using a mixture of critics as done in prior work such as D4PG [4] and PMOE [5] is missing.
4. Analysis of the potential downsides of learning mixture policies in terms of the compute requirements hasn’t been discussed, e.g., the tradeoff between wall clock time and sample complexity, etc.

[1] Baram, Nir, Guy Tennenholtz, and Shie Mannor. "Maximum entropy reinforcement learning with mixture policies." arXiv preprint arXiv:2103.10176 (2021).

[2] Plappert, Matthias, et al. "Multi-goal reinforcement learning: Challenging robotics environments and request for research." arXiv preprint arXiv:1802.09464 (2018).

[3] Rajeswaran, Aravind, et al. "Learning complex dexterous manipulation with deep reinforcement learning and demonstrations." arXiv preprint arXiv:1709.10087 (2017).

[4] Barth-Maron, Gabriel, et al. "Distributed distributional deterministic policy gradients." arXiv preprint arXiv:1804.08617 (2018).

[5] Ren, Jie, et al. "Probabilistic mixture-of-experts for efficient deep reinforcement learning." arXiv preprint arXiv:2104.09122 (2021).

**Questions:**

1. As shown in Figure 5, the Q-value estimates are noisy for sparse-reward ‘MountainCar’. Could the authors shed some light on how learning the mixture weighting policy as opposed to a fixed weighting policy as used in [1] is particularly helpful in such cases?
2. Just out of curiosity, did the authors observe any specific benefits or have a strong motivation behind training mixture policies using stochastic gradients as opposed to deterministic gradients?

[1] Baram, Nir, Guy Tennenholtz, and Shie Mannor. "Maximum entropy reinforcement learning with mixture policies." arXiv preprint arXiv:2103.10176 (2021).

---

> ### Author Response · Authors · 2024-11-21
> **Thank you for your review - I**
>
> Thank you for your effort in reviewing our paper and providing feedback. We truly appreciate your acknowledgment of our contribution. Here are our responses to your questions and concerns:
>
> **Q1**: Could the authors shed some light on how learning the mixture weighting policy as opposed to a fixed weighting policy as used in [1] is particularly helpful in such cases?
>
> **Response**: First, using a fixed weighting policy can be challenging without domain knowledge to specify the desired fixed weights. Second, a fixed weighting policy imposes restrictions on the policy class compared to allowing the weights to adapt dynamically. For instance, a uniform weighting may fail to represent certain distributions, such as those seen in the middle (Step 40k) of the bottom row in Figure 5. This limitation can be problematic, as different directions might have unequal potential for yielding rewards.
>
> Third, fixed-weight mixture policies struggle to introduce a new mode far from the existing ones. Since all components retain non-zero weights, they tend to concentrate on the current modes. To generate a new mode, one of the components must physically move to the desired new location, which can be inefficient. In contrast, mixture policies with learnable weights can seamlessly adapt by increasing the weight of a component already positioned near the desired mode, allowing for faster and more efficient adaptation.
>
> **Q2**: did the authors observe any specific benefits or have a strong motivation behind training mixture policies using stochastic gradients as opposed to deterministic gradients?
>
> **Response**: We kindly ask the reviewer to clarify what they mean by “stochastic gradients” and “deterministic gradients.” If the question pertains to the motivation for using stochastic component policies instead of deterministic ones (such as those employing deterministic policy gradients, as in DDPG (Lillicrap et al., 2016) and TD3 (Fujimoto et al., 2018)), our response is as follows: SAC maintains a stochastic policy (squashed Gaussian) and leverages the reparameterization trick for gradient computation. Therefore, using stochastic component policies aligns naturally with the baseline.
>
> Furthermore, we believe stochastic policies offer advantages in general, as they enable more flexible action distributions. In contrast, deterministic policies rely on a fixed standard deviation for exploration, which limits their adaptability. This flexibility is particularly beneficial in complex environments where diverse action distributions may be necessary for optimal performance.
>
> **Weakness 1**: The sparse reward environments investigated in the paper are simplistic. ‘Pendulum’, ‘MountainCar’, and ‘Acrobot’ are generally treated as toy domains. Experiments on more realistic sparse-reward domains such as ‘Fetch’ [2] and/or ‘Adroit’ [3] domains are missing. This limits the applicability and scalability of the proposed methods.
>
> **Response**: Thank you for pointing out these interesting domains. We tested *Pen* from [3] and *FetchReach* and *FetchSlide* from [2] because other environments are too difficult for algorithms without advanced exploration strategies. The results are presented in Appendix E.4 in the updated submission. In summary, the performance difference in Pen and FetchReach is quite small as these two environments are quite easy. In FetchSlide, one variant of mixture policies (GumbelRep-SGM) performs slightly better than the baseline, while the other variant performs slightly worse. While it’s difficult to draw a conclusion from this limited result, we leave further investigation for future work.
>
> **Weakness 2**: Comparison of the proposed methods against existing work using mixture policies such as SACM [1] in Section 5.
>
> **Response**: We answered the intuition why SACM might not be desirable because of its restriction on the mixture policies in our response to Q1. Here, we performed additional experiments to examine the performance of SACM in classic control domains with unshaped rewards. The results are presented in Appendix E.2 in the updated paper. Note that our proposed gradient estimators alleviate SACM’s (denoted as UniformRep-SGM in Figure 13) restriction on mixture policies and perform consistently better, while SACM’s performance can sometimes be worse than the base policy.
>
> **Weakness 3**: A discussion on the benefits of using mixture policies as opposed to using a mixture of critics as done in prior work such as D4PG [4] and PMOE [5] is missing.
>
> **Response**: We didn’t investigate the line of work that uses mixture distributions for the critic function because its emphasis is on distributional RL, which is orthogonal to policy parameterization for continuous control. Nevertheless, we thank the reviewer and acknowledge the connections between this line of work and our work, and we will incorporate a discussion on them in future versions.

---

> ### Author Response · Authors · 2024-11-21
> **Thank you for your review - II**
>
> **Weakness 4**: Analysis of the potential downsides of learning mixture policies in terms of the compute requirements hasn’t been discussed, e.g., the tradeoff between wall clock time and sample complexity, etc.
>
> **Response**: The computation time for training mixture policies depends on the type of the gradient estimator. In general, when both the base policy and the mixture policy don’t use a baseline, the additional computation for the mixture policy is smaller since we only need extra heads in the actor network for the mixture parameters. Further, Like-Squashed, Like-SGM, and HalfRep-SGM require a baseline to reduce the variance and stabilize learning, which makes them slower compared to those that don’t need a baseline (Rep-Squashed, GumbelRep-SGM). In our experiment, we don’t use a separate state-value (v-value) network for the baseline but compute it using the average of a batch of $30$ state-action values (q values). We included a table in Appendix D.1 in the updated paper for reference.
>
> We answered all your questions. We hope our answers can address your concerns. If you have questions or concerns, please let us know so that we can discuss them further. We appreciate your comments and the valuable time spent reading our paper and writing reviews.

---

> ### Comment · Area_Chair_f6SA · 2024-11-23
> **From AC.**
>
> Reviewer c4Ff: if possible, can you reply to the rebuttal?

---

> > ### Comment · Reviewer_c4Ff · 2024-11-26
> >
> > I thank the authors for their responses to my questions and concerns. The additional experiments presented in Appendix E partially address questions 1 and weaknesses 1, 2, and 4. However, Figure 15 (experiments in sparse-reward robotics tasks) does not show any clear trend and is inconclusive, highlighting weakness 1. I also went over the concerns of the other reviewers regarding the limited technical novelty in the theoretical analysis presented in the paper and the responses of the authors. Given the lack of conclusive results on complex sparse-reward tasks, I maintain my assessment and score.

---

### Meta-Review · Area_Chair_f6SA · 2024-12-19

**Metareview:**

In the paper, the authors study multi-modal policies (implemented with mixtures). They derive two policy gradient estimators that work with such policies. They also conduct an empirical study, finding that mixture policies do not provide a benefit in standard tasks (while providing it in tasks specially tailored to benefit).

The main strength of the paper lies in the research hypothesis being tested. Ostensibly, multi-modal policies should help so a negative result showing that they don't (at least not on standard tasks) is interesting and valuable.

The main weakness of the paper is the limited novelty (the estimators are derived using existing techniques) as well as limited empirical success.

Since most reviewers think that the paper is below bar for ICLR, I recommend rejection. I also encourage the authors to resubmit (potentially to TMLR since it seems to be more accepting of negative results).

**Additional Comments On Reviewer Discussion:**

During the discussion phase, I tired to convince the reviewers that having a negative result like that is valuable to the ICLR community, but it was impossible to change their minds. For this reason, I recommend rejection.

---

### Decision · Program_Chairs · 2025-01-22

Reject